JCB Journal of Cell Biology

# REPORT

# The distinct localization of CDC42 isoforms is responsible for their specific functions during migration

Yamini Ravichandran[1,2]*, Jan Hänisch[1]*, Kerren Murray[1]*, Vanessa Roca[1], Florent Dingli[3], Damarys Loew[3], Valentin Sabatet[3], Batiste Boëda[1], Theresia E. Stradal[4], and Sandrine Etienne-Manneville[1]

**The small G-protein CDC42 is an evolutionary conserved polarity protein and a key regulator of polarized cell functions, including directed cell migration. In vertebrates, alternative splicing gives rise to two CDC42 proteins: the ubiquitously expressed isoform (CDC42u) and the brain isoform (CDC42b), which only differ in their carboxy-terminal sequence, including the CAAX motif essential for their association with membranes. We show that these divergent sequences do not directly affect the range of CDC42's potential binding partners but indirectly influence CDC42-driven signaling by controlling the subcellular localization of the two isoforms. In astrocytes and neural precursors, which naturally express both variants, CDC42u associates with the leading-edge plasma membrane of migrating cells, where it recruits the Par6-PKCζ complex to fulfill its polarity function. In contrast, CDC42b mainly localizes to intracellular membrane compartments, where it regulates N-WASP-mediated endocytosis. Both CDC42 isoforms contribute their specific functions to promote the chemotaxis of neural precursors, demonstrating that their expression pattern is decisive for tissue-specific cell behavior.**

## Introduction

CDC42 is a small G protein of the Rho family (Etienne-Manneville and Hall, 2002; Etienne-Manneville, 2004). Like most Rho GTPases, it switches between a GDP-bound inactive status and a GTP-bound form that can activate a large panel of effector proteins. Through these effectors, RhoGTPases participate in numerous signaling cascades regulating a wide range of cellular responses. CDC42 interactions with its regulators and effectors generally occur on the cytosolic sides of cellular membranes whose planar geometry is thought to facilitate molecular interactions. Specific GEFs and GAPs regulate GTP-GDP cycling of CDC42 (Hodge and Ridley, 2016). The membrane localization of CDC42, like of other RhoGTPases, relies on the posttranslational prenylation of the cysteine included in the carboxy-terminal CAAX box. This modification promotes the protein anchorage in the lipid bilayer. GDIs can interact with the lipid anchors of Rho GTPases to sequester them in the cytosol and thereby inhibit their functions (Hodge and Ridley, 2016; Johnson et al., 2009; Nomanbhoy et al., 1999).

The Rho GTPase CDC42 is more particularly known for its fundamental role in the control of polarity during cell asymmetric division, cell differentiation, and cell migration in organisms ranging from yeast to mammals (Etienne-Manneville, 2004). CDC42–mediated signaling controls cytoskeleton rearrangement, which affects various actin and/or microtubule-dependent cellular processes and plays a key role in clathrin-dependent and clathrin-independent endocytosis, in exocytosis, and in vesicle transport (Chadda et al., 2007; Chen et al., 2005; Erickson and Cerione, 2001; Harris and Tepass, 2010; Mayor et al., 2014; Ridley, 2006; Sabharanjak et al., 2002; Wu et al., 2000). Functional dysregulation of CDC42 has been implicated in the pathology of several disease states and developmental disorders, including cancer (Aspenström, 2018; Martinelli et al., 2018). Studies using constitutively active or dominant-negative CDC42 mutants showed that CDC42 acts as an oncoprotein that promotes cellular transformation and metastasis in the context of loss of polarity (Fidyk et al., 2006; Haga and Ridley, 2016; Johnson et al., 2010; Reymond et al., 2012). On the contrary, gene knockout studies suggest that CDC42 might also function as a tumor suppressor because targeted knockout of the *CDC42* gene in hepatocytes or blood stem/progenitor cells resulted in the development of hepatocellular carcinoma or myeloproliferative disease in mice (van Hengel et al., 2008). This ambiguity makes

[1]UMR3691 CNRS, Equipe Labellisée Ligue 2023, Université de Paris, Cell Polarity, Migration and Cancer Unit, Institut Pasteur, Paris, France; [2]Collège Doctoral, Sorbonne Université, Paris, France; [3]PSL Research University, Centre de Recherche, Laboratoire de Spectrométrie de Masse Protéomique, Institut Curie, Paris, France; [4]Helmholtz Centre for Infection Research, Inhoffenstrasse 7, Braunschweig, Germany.

*Y. Ravichandran, J. Hänisch, and K. Murray contributed equally to this paper. Correspondence to Sandrine Etienne-Manneville: setienne@pasteur.fr.

understanding the role of CDC42-driven polarization in the context of cancer biology challenging.

Previous CDC42 studies have been mostly focused on the ubiquitous CDC42 isoform (CDC42u). However, the human *CDC42* gene located on chromosome 1 gives rise to three transcripts via alternative splicing, which translate into two distinct CDC42 isoforms. CDC42u, initially described as placental CDC42, does not include the exon 6 but an alternative exon 7 and is ubiquitously expressed (Marks and Kwiatkowski, 1996). In contrast, the so-called brain isoform (CDC42b) is generated by translation of the exons 1–6 and was initially detected in brain tissue. The expression of the ubiquitous isoform CDC42u in neuronal precursors and non-neuronal cells switches to a stable coexpression of the CDC42b and CDC42u isoforms. During neurogenesis, CDC42u specifically drives the formation of neuroprogenitor cells, whereas CDC42b is essential for promoting the transition of neuroprogenitor cells to neurons (Endo et al., 2020). In cortical neurons, CDC42u, whose mRNA preferentially localizes into axons, plays a role in axonogenesis (Garvalov et al., 2007; Lee et al., 2021), whereas the palmitoylation of CDC42b accounts for its preferential localization to dendritic spines and its role in dendrite maturation. These findings have advanced the understanding of mechanisms underlying axo-dendritic polarity in developing neurons and argue that coexpression of the non-redundant CDC42 isoforms is important during neuronal development (Yap et al., 2016). However, the molecular mechanisms responsible for the specific functions of these two isoforms as well as their role in non-neuronal cells remain unknown. In fact, both isoforms were found to be expressed in a range of commonly used laboratory cell lines, including HEK and MDCKII cells (Wirth et al., 2013). This splice variant may thus also be expressed in non-brain tissue cells underlining the need to clarify the functional differences between the two CDC42 variants in non-cell specific functions. By inactivating specifically each of the two CDC42 variants in brain cells such as astrocytes and neural precursor cells (NPC), we unravel the specific localization and functions of the two isoforms during persistent directed migration.

## Results

### Cell polarization relies on ubiquitous CDC42

To determine whether CDC42 isoforms had different functions, we used primary astrocytes and glial cells in which we determined the expression levels of CDC42 isoforms. Primary rat astrocytes were transfected with siRNAs designed to selectively knockdown brain (si-b1, si-b2) or ubiquitous (si-u1, si-u2) CDC42 or with an siRNA targeting a common sequence (si-both) to simultaneously inhibit both isoforms. Due to the absence of isoform-specific antibodies, we used qPCR to quantify CDC42u and total CDC42 mRNA levels. Knockdown of each isoform revealed that roughly 15–20% of total CDC42 mRNA in astrocytes encodes the brain isoform (Fig. S1, A and B) (Yap et al., 2016).

We then chose to determine the contribution of each CDC42 isoform in cell-directed migration, a more complex and integrated cell function. Control and CDC42 knockdown cells were submitted to a scratch-induced migration assay, a well-characterized model system in which we have previously demonstrated that CDC42 is activated by integrin-mediated signaling and involved in microtubule-dependent front-rear cell polarization, Golgi and centrosome reorientation, and directed membrane recycling (Etienne-Manneville and Hall, 2001; Osmani et al., 2006). Knockdown of both CDC42 isoforms did not influence the migration velocity but led to a strong decrease in directionality (83% to about 55%) and persistence (85% to about 60%) of migration (Fig. 1, A–C). The specific depletion of CDC42u led to similar results with a significant decrease in the directionality and persistence of migration, whereas CDC42b-depleted cells migrated similarly as control astrocytes (Fig. 1, A–C).

Associated with the alteration of directional persistence, the number of filopodia, the classic actin-driven function of CDC42 at the cell leading edge was strongly inhibited by the simultaneous depletion of both isoforms. A similar result was observed upon specific depletion of CDC42u. In contrast, CDC42b depletion alone had no impact on filopodia formation, suggesting a more predominant role of CDC42u in this phenomenon (Fig. S1 C). To assess the mechanisms involved in the control of directed migration, we looked at Golgi reorientation in cells at the wound edge. Following the wounding of the astrocyte monolayer, the Golgi apparatus together with the centrosome localizes in front of the nucleus in the direction of migration in a CDC42-dependent manner (Etienne-Manneville and Hall, 2001). Here, microtubule polarized rearrangement and Golgi reorientation were dramatically impaired by the depletion of both CDC42 isoforms as expected and were similarly affected by the specific depletion of the CDC42u (Fig. 1, D and E). In contrast, CDC42b knockdown had no detectable effect (Fig. 1, D and E). The Par6/PKCζ polarity complex is a key mediator of CDC42 function in astrocyte polarization (Etienne-Manneville and Hall, 2001, 2003a, 2003b; Etienne-Manneville et al., 2005). Depletion of CDC42u alone or of both CDC42 variants, but not of CDC42b alone, prevented the recruitment of the aPKC protein family member PKCζ to the cell leading edge (Fig. 1, F and G), confirming that CDC42u, but not CDC42b, is required for the activation of the polarity pathway in astrocytes. To determine whether the lack of function of CDC42b was due to its relatively low level of expression, siRNA-resistant CDC42u (u$^{RES}$) or CDC42b (b$^{RES}$) were expressed. u$^{RES}$, but not b$^{RES}$, rescued Golgi reorientation in CDC42-depleted astrocytes (si-both) (Fig. 1 H). Together, these observations point to the ubiquitous CDC42u isoform as the main regulator of cell polarity and directed migration in astrocytes.

### Brain and ubiquitous isoforms show similar binding partners

We then investigated the molecular differences that could explain the functional specificity of the two CDC42 isoforms. The alternative splicing of the carboxy-terminal (C-ter) exons causes a major difference in the last 10 amino acids of CDC42 isoforms (Fig. 2 A). Since the C-ter domain of Rac was shown to participate in its interaction with some of its effectors (Abdrabou and Wang, 2018; Knaus et al., 1998), we examined whether both CDC42 variants could interact with the same effectors. We first tested their interaction with Par6 and PKCζ by performing

Figure 1. **CDC42u, but not CDC42b, controls cell polarization and directed and persistent migration of astrocytes. (A)** Representative trajectories of astrocytes transfected with siRNAs against the ubiquitous (si-u1), the brain (si-b1), or both (si-both) CDC42 isoforms and migrating in an in vitro wound healing

assay for 16 h. **(B and C)** Directionality (B) and directional persistence (C) of astrocytes transfected with the indicated siRNA and migrating in a scratch wound assay. **(D)** Immunofluorescence images of wound-edge astrocytes transfected with the indicated siRNA and fixed 8 h after wounding. Images show microtubules (anti-tubulin, white), cis-Golgi (anti-GM130, green), centrosome (anti-pericentrine, red), and the nucleus (DAPI, blue). **(E)** Quantification of Golgi orientation in astrocytes transfected with the indicated siRNA, 8 h after wounding. The red line indicates the values expected for random positioning of the Golgi apparatus. **(F)** Fluorescence images showing PKCζ localization in wound-edge astrocytes transfected with the indicated siRNA. **(G)** Percentage of wound-edge cells showing PKCζ accumulation at the cell front. **(H)** Quantification of Golgi reorientation in a rescue experiment using astrocytes transfected with control siRNA or with a siRNA targeting both CDC42 isoforms together with the indicated control (GFP) or GFP-tagged CDC42 constructs. Graphs show data presented as means ± SEM of 5 (C), 4 (E and G), and 3 (H) independent experiments. At least 250 cells (A–C) and 150 cells (D–H) were analyzed per condition. Ctl: Control cells transfected with non-relevant siRNA. P values were calculated using two-sided unpaired Student's *t* test. Scale bars: 10 µm.

immunoprecipitation of constitutively active (CA) V12-, GFP-tagged CDC42b, and CDC42u in HEK cells (Fig. 2 B). In these conditions, both isoforms appear to interact similarly with the Par6-aPKCζ polarity complex (Fig. 2 C).

Thus, we next performed GFP-trap pull-down assays using HEK cells overexpressing GFP-tagged CDC42 isoforms (Fig. 2 D). The GFP-tagged CA-CDC42 resins along with coimmunoprecipitated interactors were sent for a mass spectrometric proteomic screen. Quantitative analysis was performed to assess the fold change of peptides bound to one isoform of CDC42 in comparison with GFP control. Analysis of best binding partners was performed with the CA mutant of each isoform to capture a maximum of effector proteins (see Materials and methods for access to PRIDE repository containing raw data and screen analysis). We then performed loose filtering parameters to segregate the interactors of the screen into effectors and also GEFs and GAPs. We identified 29 effectors out of 40 known epithelial effector proteins of CDC42 (Fig. 2 D and Fig. S2 A) (Pichaud et al., 2019). We could also detect seven known GEFs (Fig. 2 E and Fig. S2 B) and five known GAPs (Fig. 2 F and Fig. S2 C). We could not identify any statistically significant difference in the list of effectors or in the fold change of effectors between both isoforms (Fig. S2 and PRIDE repository). These results show that despite their divergent carboxy-terminal sequences, both CDC42 isoforms can biochemically interact with the same binding partners. From the correlation plots of the corresponding fold change for each isoform, we analyzed the linear regression for all three plots. $R^2 = 0.893$ for GEFs, 0.900 for GAPs, and 0.978 for effectors, therefore indicating a good linear fit correlation (Fig. 2, D–F), which may suggest that the alternative splicing does not strongly modify the binding affinity of the CDC42 with its main partners.

### Brain and ubiquitous CDC42 display different intracellular localization

Seeking an alternative explanation to the functional specificity of the two isoforms, we asked whether the carboxy-terminal sequence, which includes the CAAX box required for membrane recruitment of the CDC42 proteins, may influence the protein subcellular localization and ability to encounter its interactors. Immunoprecipitation experiments are typically performed using cells that strongly overexpress the tagged CDC42 (Fig. 2 G), leaving the possibility that a specific subcellular localization of endogenous protein isoforms may be a critical factor in the control of CDC42 interactions with its effectors.

Since we could not find nor produce any antibody that specifically recognizes CDC42 isoforms in immunoprecipitation or in immunofluorescence, we microinjected constructs encoding fluorescently tagged CDC42 isoforms into astrocyte nuclei. Cells were imaged after a short time of expression (4 h) to obtain low levels of expression. In contrast to cell transfection, short-term expression after microinjection revealed a different localization of the two CDC42 isoforms in astrocytes (Fig. 3 A; and Videos 1, 2, and 3). Video 1 shows non-migrating astrocytes expressing GFP-CDC42b together with mCherry-CDC42u. CDC42u is mainly cytosolic, whereas CDC42b appears to be largely associated with intracellular compartments. Videos 2 and 3 show the localization of CDC42 isoforms in cells actively migrating in a scratch-induced migration assay, previously shown to activate CDC42 through the integrin-mediated recruitment of the GEF βPIX (Osmani et al., 2010). CDC42u accumulated at the leading edge plasma membrane, including in filopodia. CDC42b was recruited to the cell front, generally slightly behind the leading edge (Fig. 3 A). CDC42b was most strikingly found on intracellular membranes, including cytoplasmic vesicles, where it colocalized with the early endosome marker EEA1, and Golgi apparatus, where it colocalized with the cis-marker GM130 (Fig. 3, A–C). In contrast, CDC42u was rarely detectable at these sites (Video 3 and Fig. 3, A–C). This differential subcellular localization of the isoforms was confirmed using a different, smaller ALFA tag (Fig. S1 D).

We asked whether specific subcellular localization was due to the divergent C-ter domain of CDC42. The C-ter CAAX box of both isoforms is geranyl-geranylated; however, CDC42b does not always undergo proteolysis and methylation after genranyl-geranylation and can bear an additional palmitoyl group attached to the last cysteine residue in a reversible manner (Kang et al., 2008; Nishimura and Linder, 2013, 2019). Suppression of the geranyl-geranylation by a CVLL to SVLL mutation in CDC42u (u[SVLL]) (Nishimura and Linder, 2013) led to its accumulation in the nucleus (Fig. 3 D). A similar CCIF to SSIF mutation in CDC42b (b[SSIF]) to prevent all lipid modification of CDC42b resulted in CDC42b's homogenous distribution across the nucleus and cytoplasm and complete loss of endomembrane-binding and cytosolic distribution of the protein. A lower nuclear accumulation was shown by b(SCIF), yet still exhibiting loss of endomembrane-binding and cytosolic distribution of the protein. In contrast, the CCIF to CSIF mutation, which specifically prevents palmitoylation but not geranyl-geranylation did not prevent its association with the Golgi apparatus (Fig. 3 D), suggesting that palmitoylation controls the recruitment of CDC42b to the cytoplasmic vesicles from the Golgi apparatus. We confirmed the role of lipid anchors in the membrane association of CDC42 isoforms by treating cells with GGTI298, which prevents both geranyl-geranylation and palmitoylation,

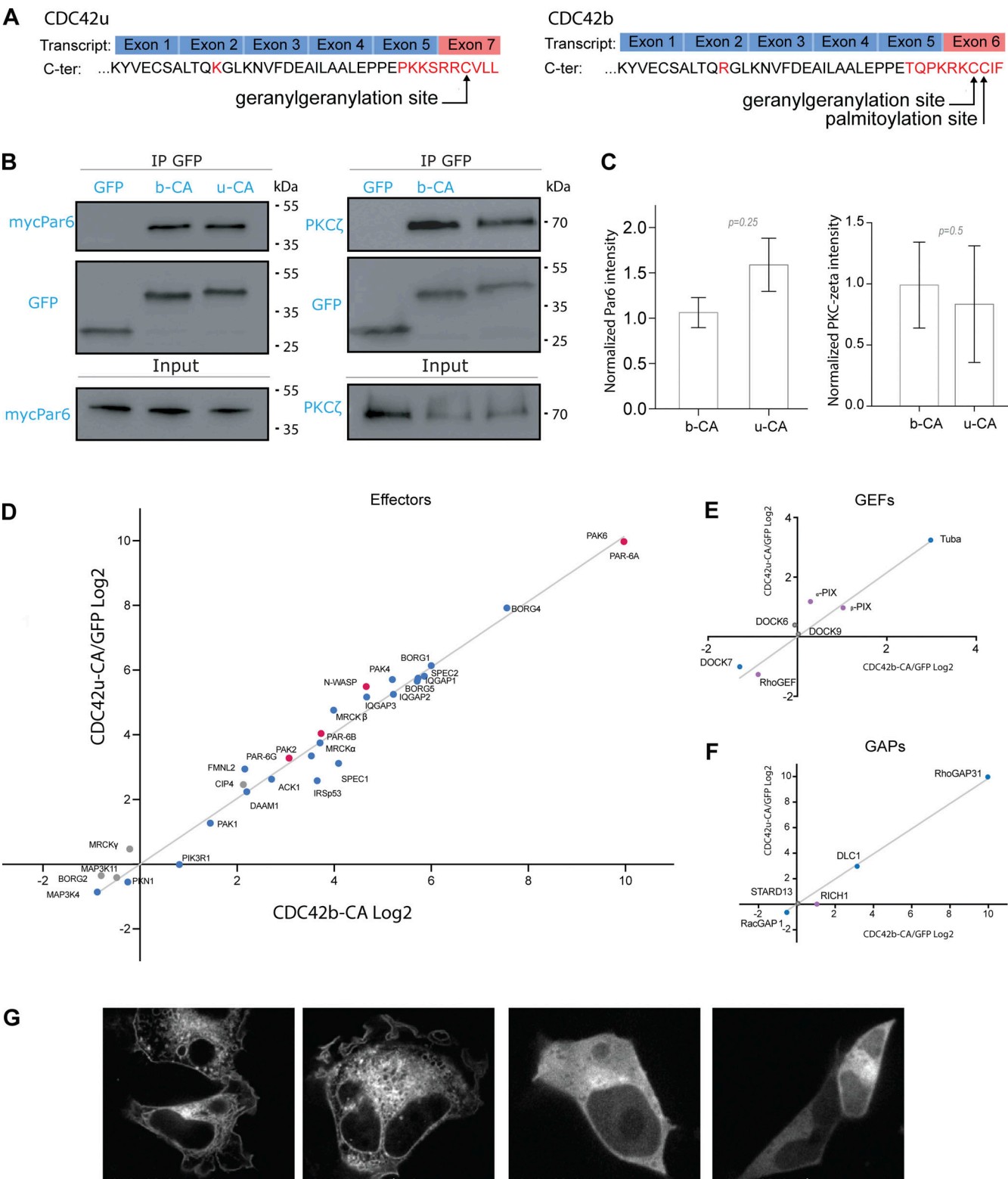

Figure 2. **CDC42u and CDC42b share the same panel of binding partners. (A)** Transcribed exons and translated carboxy-terminal protein sequences of ubiquitous (CDC42u) and brain CDC42 (CDC42b). **(B)** Western blots showing immunoprecipitation of GFP-tagged proteins; GFP-CDC42b CA (b-CA) and GFP-CDC42u CA (u-CA) from transfected HEK cells. Coimmunoprecipitation of mycPar6 and PKCζ are shown in the top panels and their respective expression levels in total cell lysate in the lower input panel. **(C)** Quantification of coimmunoprecipitated mycPar6 and PKCζ normalized to their input values. The graph shows data points and means ± SEM of three independent experiments. **(D–F)** Correlation analysis plots showing the fold change (in Log$_2$ units) in the number of peptides of a given protein coprecipitated with constitutively active (CA) CDC42b (x-axis) versus with CDC42u (y-axis). The number of peptides is normalized by the respective number obtained in GFP control immunoprecipitation. Proteomic screening was performed by applying loose filtering parameters to

segregate the interactors of the CA screen into GEFs, GAPs, and effectors. We identified 29 effectors out of 40 known epithelial effector proteins of CDC42. N-WASP and PAR6 proteins have been highlighted in red (E), seven known GEFs (F), and five known GAPs (G). **(D–F)** Peptides used to calculate significance are ≥6, P value ≤0.05, and number of replicates = 4. **(G)** Spinning disk images of HEK cells overexpressing the indicated pEGFP-CDC42 constructs 24 h after transfection. Note that, in these conditions, CDC42u and CDC42b display similar localization pattern. Source data are available for this figure: SourceData F2.

or 2-bromopalmitate (2BP), a specific inhibitor of palmitoylation (Fig. S3 A).

These results show that the lipid modifications of the carboxy-terminal domain of CDC42 isoforms are crucial for the specific membrane association of the two proteins. They further suggest that the divergent C-ter domains may contribute to the functional specificities of CDC42 isoforms by controlling their subcellular localization, indirectly affecting their ability to interact with their effectors. Moreover, the fact that palmitoylation as well as scratch-induced signaling influence CDC42b recruitment to trafficking vesicles from the Golgi apparatus suggests that each isoform may be regulated by distinct pathways. The fact that GDI3, a GDI expressed in the brain and pancreas, localizes at the Golgi apparatus via an N-terminal hydrophobic anchor (Brunet et al., 2002) suggests yet another potential regulatory mechanism.

**Brain CDC42 is the major CDC42 isoform involved in N-WASP-dependent endocytosis**

Following this line of reasoning, we sought CDC42b-specific functions on intracellular membrane compartments. In migrating astrocytes, CDC42 not only controls Par6-dependent cell polarity but is also involved together with Arf6 in vesicular recycling from the plasma membrane (Osmani et al., 2010). Because CDC42b showed a strong enrichment on EEA1-positive organelles compared with CDC42u (Fig. 3, A–C), we performed dextran uptake experiments to examine the role of CDC42 isoforms in the formation of the EEA1-positive pinosomes, which are frequently observed at the leading edge of migrating astrocytes. Knockdown of both isoforms of CDC42 strongly reduced (–47%) dextran uptake (Fig. 4 A). Strikingly, CDC42u depletion caused a minor reduction of dextran internalization, whereas CDC42b knockdown significantly decreased the uptake rates (by ∼40%; Fig. 4 A). The predominant role of CDC42b in pinocytosis was confirmed by rescue experiments in astrocytes depleted for both isoforms. GFP-CDC42b[RES] led to a stronger rescue than GFP-CDC42u[RES] (Fig. 4 B). The non-lipid-modified mutants of either isoform (b[RES][SCIF], u[RES][SVLL]) did not rescue dextran uptake (Fig. 4 B). Furthermore, overexpression of the non-palmitoylable CDC42b mutant (b[RES][CSIF]) did not restore the dextran uptake (Fig. 4 B), indicating that palmitoylation, which promotes CDC42b association with intracellular vesicles specifically (Fig. 3 D), is crucial for its function in pinocytosis. Dextran uptake experiments using specific inhibitors of lipid modification (GGTI298 or 2BP) confirmed these findings (Fig. S3 B). These results show that palmitoylated CDC42b is the major CDC42 isoform involved in pinocytosis in migrating astrocytes.

siRNA-mediated depletion of N-WASP (Fig. 4 C) inhibited dextran uptake in migrating astrocytes to a similar level as CDC42b knock-down (Fig. 4, A and B) as previously reported

(Kessels and Qualmann, 2002; Legg et al., 2007). A double knockdown of CDC42b and N-WASP did not further increase the reduction in dextran uptake (∼40%) (Fig. 4 D). Finally, expression of an activated form of N-WASP (GFP-NWCA) in CDC42b-depleted cells rescued macropinocytosis, confirming that N-WASP is the major effector of CDC42b controlling macropinocytosis in astrocytes (Fig. 4 E). We looked at the localization CDC42b colocalized with N-WASP on macropinosomes in migrating astrocytes, whereas CDC42u was rarely observed on any intracellular vesicles including macropinosomes (Fig. 3, A and C; and Fig. 4 F and Video 4). These data indicate that the C-ter domain of CDC42b controls its association with intracellular vesicles and its interaction with N-WASP- and CDC42/N-WASP-mediated pinocytosis. However, when N-WASP association with CDC42 isoforms was assessed in overexpressing HEK cells (Fig. 4 G), there was no significant difference in the ability of phospho-N-WASP to bind each isoform (Fig. 4, G and H). The observation that both CDC42 isoforms predominantly pull down phospho-N-WASP indicates their comparable ability to relieve N-WASP autoinhibition, thereby triggering the subsequent tyrosine phosphorylation of N-WASP by members of the Src kinase family (Bompard and Caron, 2004). These results are reminiscent of the one observed in proteomic analysis of CDC42 interactome (Fig. 2 E). We conclude that even if both isoforms can interact with N-WASP in vitro, CDC42b is the main regulator of N-WASP-dependent micropinocytosis in cells because of its specific localization on endocytic vesicles. More generally our findings illustrate the importance of subcellular compartmentalization in CDC42 and more generally Rho GTPases' functions. The control of CDC42 localization by its carboxy-terminal sequence and by lipid modifications (Farhan and Hsu, 2016; Ravichandran et al., 2020) may explain how mutations that touch this carboxy-terminal variable sequence of CDC42 and affect either one or both CDC42 are associated with distinct dramatic disorders (Bekhouche et al., 2020; Martinelli et al., 2018).

**Both CDC42 isoforms contribute their specific functions during chemotaxis of neural precursor cells**

Since CDC42 isoforms show functional differences, we asked whether they may cooperate in more complex migratory situations where both front–rear polarization and endocytosis are required. Neural precursor cells (NPCs) migrate long distances from their zones of origin to their final destination where they differentiate, following gradients of chemoattractants (Leong et al., 2011). In NPCs, both isoforms are expressed at approximately identical levels (Fig. 5 A) (Yap et al., 2016). Like in astrocytes, short-term expression of microinjected CDC42 constructs in NPCs illustrated the distinct subcellular localization of CDC42u and CDC42b. CDC42u was mainly visible in the cytosol and at the plasma membrane, and the CDC42b

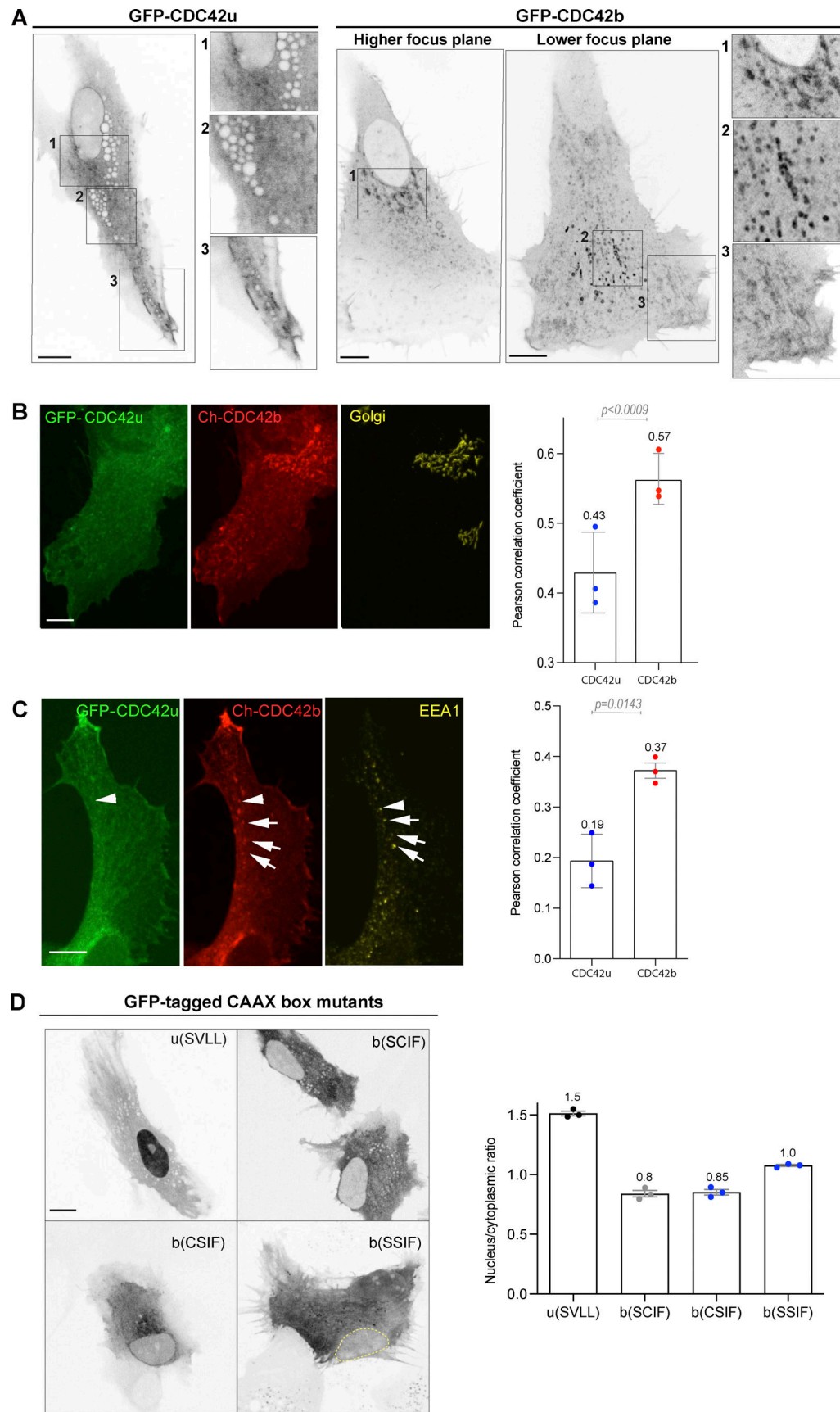

Figure 3. **Localization of brain CDC42 (CDC42b) and ubiquitous CDC42 (CDC42u) in migrating astrocytes. (A)** Confocal section images of migrating astrocytes 3–4 h following microinjection of GFP-tagged CDC42u or CDC42b constructs and 5 h after wounding. The right panels show higher magnification of

the corresponding boxed area and highlight the different localization of the two isoforms (see corresponding Videos 2 and 3). **(B)** Confocal stack images of GFP-CDC42u- and mCherry-CDC42b-expressing astrocytes 5 h after wounding stained with anti-GM130 (cis-Golgi marker). Right panel, quantification of the colocalization of each CDC42 construct with GM130. **(C)** Confocal stack images of GFP-CDC42u and mCherry-CDC42b expressing astrocytes 5 h after wounding stained with anti-EEA1 (early endosomes marker). Right panel, quantification of the colocalization of each CDC42 construct with EEA1. **(D)** Spinning disk section images showing localization of GFP-tagged non-lipid-modified CDC42u (u[SVLL]), non-prenylated brain CDC42 (b[SCIF]), non-palmitoylatable CDC42b (bCSIF), and/or non-lipid modified CDC42b (bSSIF) in migrating astrocytes 5 h after wounding. Right panel, showing the quantification of nucleus/cytoplasmic ratios for each lipid mutant. All graphs show the values and means ± SEM of three independent experiments, and at least 30 cells were analyzed per condition. All P values were calculated using two-sided unpaired Student's *t* test. Scale bars: 10 μm.

accumulated on intracellular EEA1-positive vesicles and the Golgi apparatus (Fig. 5, B and C).

NPCs grow in a primary 3D tissue culture system (neurospheres), and their migration can be observed when neurospheres are placed on an adhesive substrate (Durbec et al., 2008) (Fig. 5 D). Using video microscopy, we first analyzed NPC migration out from neurospheres. The directional persistence was strongly reduced in CDC42u-depleted cells (from 76% to 58–60%) and in cells lacking both isoforms (to 50%), but was not significantly altered by CDC42b or N-WASP depletion (Fig. 5, D and E; and Fig. S3 C).

We next performed dextran uptake assays in siRNA-treated NPCs. CDC42b specific depletion, like N-WASP depletion, significantly decreased dextran uptake into NPCs, whereas knockdown of CDC42u had no significant effect (Fig. 5 F and Fig. S3 D). These results in NPCs confirmed the predominant role of CDC42u in cell polarization and of CDC42b in macropinocytosis.

We then tested the contribution of the two isoforms during NPC chemotactic migration, during which endocytosis is involved in the processing of chemotactic signals (Zhou et al., 2007). We used a Boyden chamber-based xCelligence assay to analyze the chemotactic migration of astrocytes and NPCs. In the absence of a gradient, astrocytes and NPCs barely migrated through the filter (Fig. S3, E and F). The addition of FBS in the lower compartment induced the chemotactic migration of both astrocytes and NPCs (Fig. 5, G and H). The knockdown of N-WASP decreased chemotaxis efficiency in NPCs but did not affect astrocyte chemotaxis (Fig. 5, G and H). When astrocytes or NPCs were transfected with siRNAs targeting both CDC42 isoforms, chemotaxis was strongly reduced in both cell types (Fig. 5, I and J). For both cell types, CDC42u-specific depletion also strongly reduced chemotactic migration. In contrast, CDC42b depletion led to a significant decrease in N-WASP-dependent NPC chemotaxis without altering N-WASP-independent astrocyte migration (Fig. 5, I and J). We concluded that CDC42b is the major isoform involved in N-WASP dependent function in NPC chemotaxis and that both isoforms of CDC42 contribute their specific functions to participate in complex migratory processes such as NPC chemotaxis.

In conclusion, we show that while both CDC42 isoforms can in principle interact with the same effectors in vitro, they do have non-redundant functions in cells. With the role of CDC42b in the formation of dendritic spines during neuronal differentiation (Kang et al., 2008; Wirth et al., 2013), our identification of its function in endocytosis and NPC chemotaxis is increasing evidence that the brain variant controls specific neural functions. NPC migration is a crucial step in brain development and conditional deletion of both CDC42 isoforms in mouse NPCs has

been shown to cause malformations in the brain (Chen et al., 2006). Nevertheless, both CDC42 isoforms contribute, albeit differently, to the behavior of NPCs, underlining that the coexpression of these non-redundant isoforms is essential during neuronal development.

# Materials and methods
## Antibodies and inhibitors
The following primary antibodies were used in this study: rat monoclonal anti-α-tubulin (MCA77G; AbDSerotec), mouse monoclonal anti-GM130 (610823; BD Transduction), rabbit polyclonal anti-pericentrin (432-C; Covance PRB), rabbit polyclonal antibodies against either pan-N-WASP or phospho-N-WASP (4848; Cell Signalling, AB23394; Abcam), rabbit polyclonal anti-PKCζ C-20 (SC-216; Santa Cruz Biotech), mouse anti-EEA1 (610457; BD biosciences), and HRP coupled anti-GFP (ab6663; Abcam). As secondary antibodies, we used standard antibodies from Jackson ImmunoResearch: Cy5 conjugated donkey anti-mouse, Alexa Fluor 488 conjugated donkey anti-rat, TRITC conjugated donkey anti-rabbit, as well as HRP coupled donkey anti-mouse, anti-rabbit, and anti-goat. DAPI in ProLong Gold Antifade Reagent (Life Tech) was used to visualize nuclei. To suppress lipid modification of CDC42 isoforms, cells were treated overnight in 120 μM 2BP (to suppress palmitoylation; Sigma-Aldrich) or 20 μM GGTI298 (to suppress geranyl-geranylation and palmitoylation; Tocris).

## Cell culture
All procedures were performed in accordance with the guidelines approved by the French Ministry of Agriculture, following European standards. Preparation of neurosphere cultures was performed as described (Calaora et al., 2001). Briefly, the striata of E14 OFA rats were removed from the embryos and mechanically dissociated before cells were seeded at 1.2 × 10⁵ cells/ml in uncoated 260-ml culture flasks (Fisher Bioblock Sc.). The culture medium consisted of DMEM/F-12 (Gibco) supplemented with 2% B27 (Gibco) and 50 μg/ml gentamicin (Sigma-Aldrich) in the presence of 20 ng/ml EGF (R&D Systems). Media were supplemented with 20 ng/ml EGF every 48 h and spheres were passaged using 0.025% trypsin-EDTA (Gibco) on the fourth and sixth day in culture. Human FGF-b (RayBiotech) was also added to the medium at 10 ng/ml for the first 4 days of culture. For the preparation of primary astrocyte cultures, the telencephala of E18 OFA rats were removed from the embryos and mechanically dissociated. Cells were plated and maintained as previously described (Etienne-Manneville, 2006) using 1g/l glucose DMEM (Gibco) supplemented with 10% FBS (Eurobio) and penicillin/

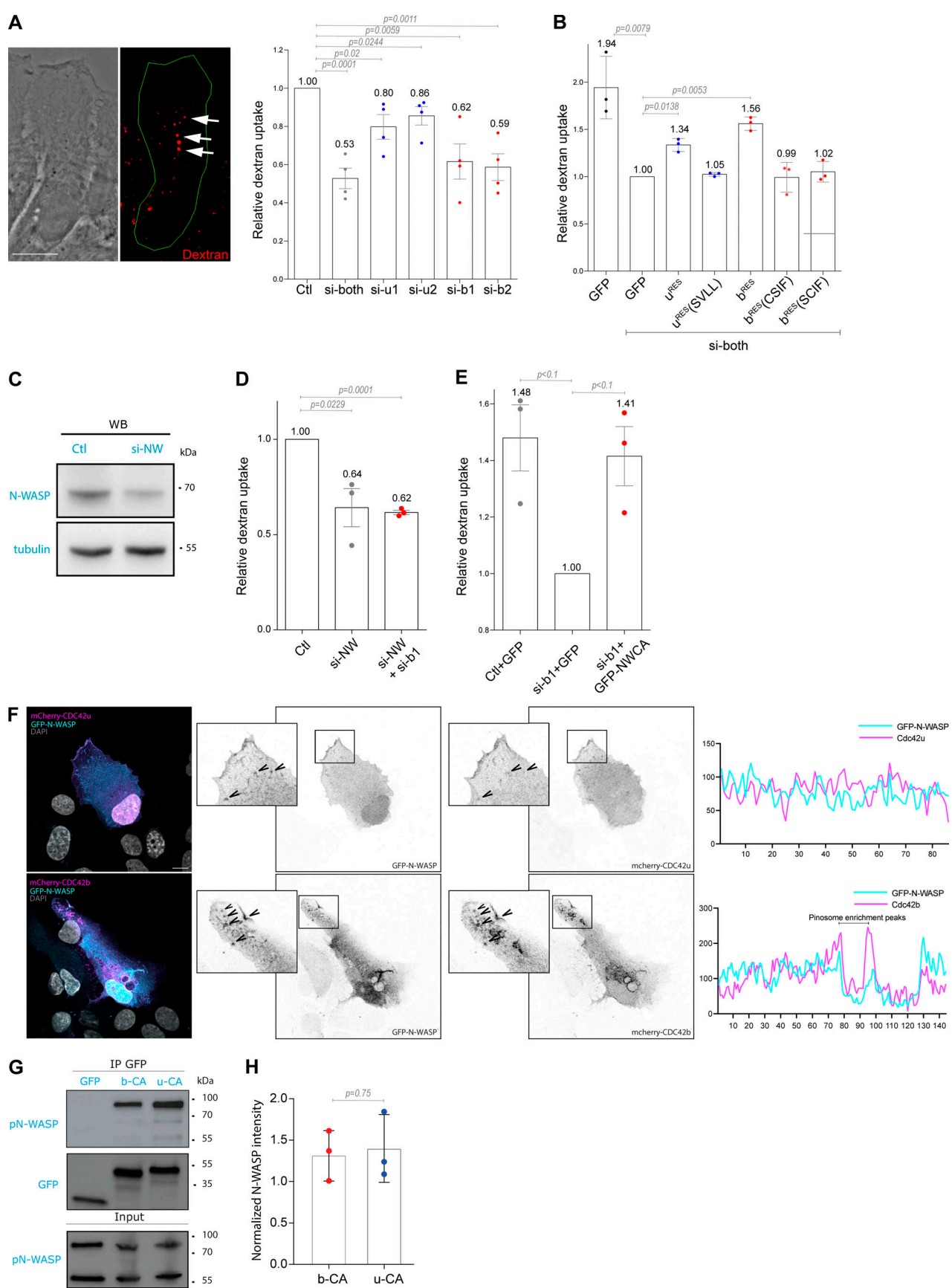

Figure 4. **CDC42b controls pinocytosis via N-WASP. (A)** Phase and fluorescence images showing dextran uptake in control migrating astrocytes. Right panel, quantification of the relative dextran uptake in astrocytes nucleofected with the indicated CDC42 siRNA. **(B)** Quantification of the relative dextran

uptake in a rescue experiment using control or CDC42-depleted astrocytes expressing the indicated CDC42 construct. **(C)** Western blot analysis of N-WASP expression in control (Ctl) and N-WASP siRNA nucleofected astrocytes. Tubulin was used as the loading control. **(D)** Quantification of the relative dextran uptake in astrocytes nucleofected with indicated siRNAs (NW: N-WASP). **(E)** Quantification of relative dextran uptake in a rescue experiment using control or CDC42b-depleted astrocytes expressing GFP or GFP-N-WCA (constitutively active N-WASP) constructs. All graphs show the values and means ± SEM of three independent experiments with at least 150 cells analyzed per condition. Data were normalized to the values obtained for si-both treated and GFP-expressing cells. **(F)** Confocal section images from migrating astrocytes expressing mCherry-tagged CDC42 isoforms and GFP-N-WASP 6h after wounding. Insets highlight the colocalization of N-WASP with CDC42b, but not CDC42u, on macropinosomes. The right panels show the corresponding fluorescence intensity profile across this region. **(G)** Western blot showing immunoprecipitation of GFP-tagged proteins; GFP-CDC42b CA (b-CA) and GFP-CDC42u CA (u-CA) from transfected HEK cells and coimmunoprecipitation of N-WASP. **(H)** Quantifications of coimmunoprecipitated N-WASP normalized to its respective expression levels in total cell lysate (input). All P values were calculated using two-sided unpaired Student's *t* test. Scale bars: 10 µm. Source data are available for this figure: SourceData F4.

streptomycin (10,000 U ml$^{-1}$ and 10,000 µg ml$^{-1}$; Gibco) as culture medium. HEK and HeLa cells were cultured in 4.5 g/l glucose DMEM (Gibco) supplemented with FBS and antibiotics as added for astrocytes. All cells were kept in a 37°C incubator at 5% $CO_2$.

### Immunoprecipitation assay and mass spectrometric proteomic screen

GFP immunoprecipitation assay was carried out, where HEK cells transfected with GFP tagged constructs of CDC42 were lysed using 50 mM TRIS base, Triton 2%, 200 mM NaCl as well as 1 tablet/10 ml protease inhibitor Mini-complete, EDTA-free (Roche). After removal of insoluble fragments via centrifugation at 12,000 *g* for 25 min, lysates were incubated with 15 µl of GFP-Trap Agarose beads from Chromotek for 1 h at 4°C on a rotary wheel. The beads were washed using a wash buffer comprising 50 mM TRIS base, 150 mM NaCl, 1 mM EDTA, and 2.5 mM $MgCl_2$ with pH adjusted to 7.5. Following the final wash, beads were stored with wash buffer at 4°C prior to depositing at the Institut Curie Mass Spectrometry and Proteomics facility (LSMP), where proteins on beads were washed twice with 100 µl of 25 mM $NH_4HCO_3$. We then performed on-beads digestion with 0.2 µg of trypsin/LysC (Promega) for 1 h in 100 µl of 25 mM $NH_4HCO_3$. The sample was then loaded onto a homemade C18 StageTips for desalting. Peptides were eluted using 40/60 MeCN/$H_2O$ + 0.1% formic acid and vacuum-concentrated to dryness.

Online chromatography was performed with an RSLCnano system (Ultimate 3000; Thermo Fisher Scientific) coupled to an Orbitrap Fusion Tribrid mass spectrometer (Thermo Fisher Scientific). Peptides were trapped on a C18 column (75 µm inner diameter × 2 cm; nanoViper Acclaim PepMapTM 100; Thermo Fisher Scientific) with buffer A (2/98 MeCN/$H_2O$ in 0.1% formic acid) at a flow rate of 4.0 µl/min over 4 min. Separation was performed on a 50 cm × 75 µm C18 column (nanoViper Acclaim PepMapTM RSLC, 2 µm, 100 Å; Thermo Fisher Scientific) regulated to a temperature of 55°C with a linear gradient of 5–25% buffer B (100% MeCN in 0.1% formic acid) at a flow rate of 300 nl/min over 100 min. Full-scan MS was acquired in the Orbitrap analyzer with a resolution set to 120,000, and ions from each full scan were HCD=fragmented and analyzed in the linear ion trap.

For identification, the data were searched against the *Homo sapiens* (UP000005640) SwissProt database using Sequest HF through proteome discoverer (version 2.2). Enzyme specificity was set to trypsin, and a maximum of two missed cleavage sites were allowed. Oxidized methionine, N-terminal acetylation, and

carbamidomethyl cysteine were set as variable modifications. The maximum allowed mass deviation was set to 10 ppm for monoisotopic precursor ions and 0.6 Da for MS/MS peaks.

The resulting files were further processed using myProMS (Poullet et al., 2007) v3.6 (work in progress). FDR calculation used a Percolator and was set to 1% at the peptide level for the whole study. The label-free quantification was performed by peptide Extracted Ion Chromatograms (XICs) computed with MassChroQ version 2.2 (Valot et al., 2011). For protein quantification, XICs from proteotypic peptides shared between compared conditions (TopN matching) with no missed cleavages were used. Median and scale normalization were applied to the total signal to correct the XICs for each biological replicate. To estimate the significance of the change in protein abundance, a linear model (adjusted on peptides and biological replicates) was performed and P values were adjusted with a Benjamini–Hochberg FDR procedure with a control threshold set to 0.05.

The mass spectrometry proteomics data have been deposited to the ProteomeXchange Consortium via the PRIDE (Vizcaíno et al., 2016) partner repository with the dataset identifier PXD017477.

### Cell transfection and RNAi

siRNA constructs were introduced into rat astrocytes or NPCs by Nucleofection Technology (Amaxa Biosystems) using Lonza protocols. Plasmids encoding fluorescently tagged constructs were microinjected. All siRNAs were obtained from Eurofins except for a non-targeting control, which was obtained from Dharmacon. To quantify knockdowns, protein samples were analyzed using ECL immunoblotting and ImageJ. Alternatively, mRNA levels were measured using qPCR (see below). In each case, cells were analyzed 4 d after transfection with siRNAs. Transfection of HEK293 and HeLa cells with plasmids was performed with the calcium phosphate method.

si-N-WASP: 5′-CUUGUCAAGUAGCUCUUAA(dTdT)-3′
si-both: 5′-UGAUGGUGCUGCUUGGUAAA(dTdT)-3′
si-u1: 5′-CAAUAAUGACAGACGACCU(dTdT)-3′
si-u2: 5′-GCAAUAUUGGCUGCCUUGGUU(dTdT)-3′
si-b1: 5′-CCAUUUAACAAUCGACUUA(dTdT)-3′
si-b2: 5′-ACUCAACCCAAAAGGAAGUUU(dTdT)-3′

### Real time qPCR

For isolation of total RNA striatal neurospheres or cultured astrocytes were prepared from E14 or E18 rat embryos,

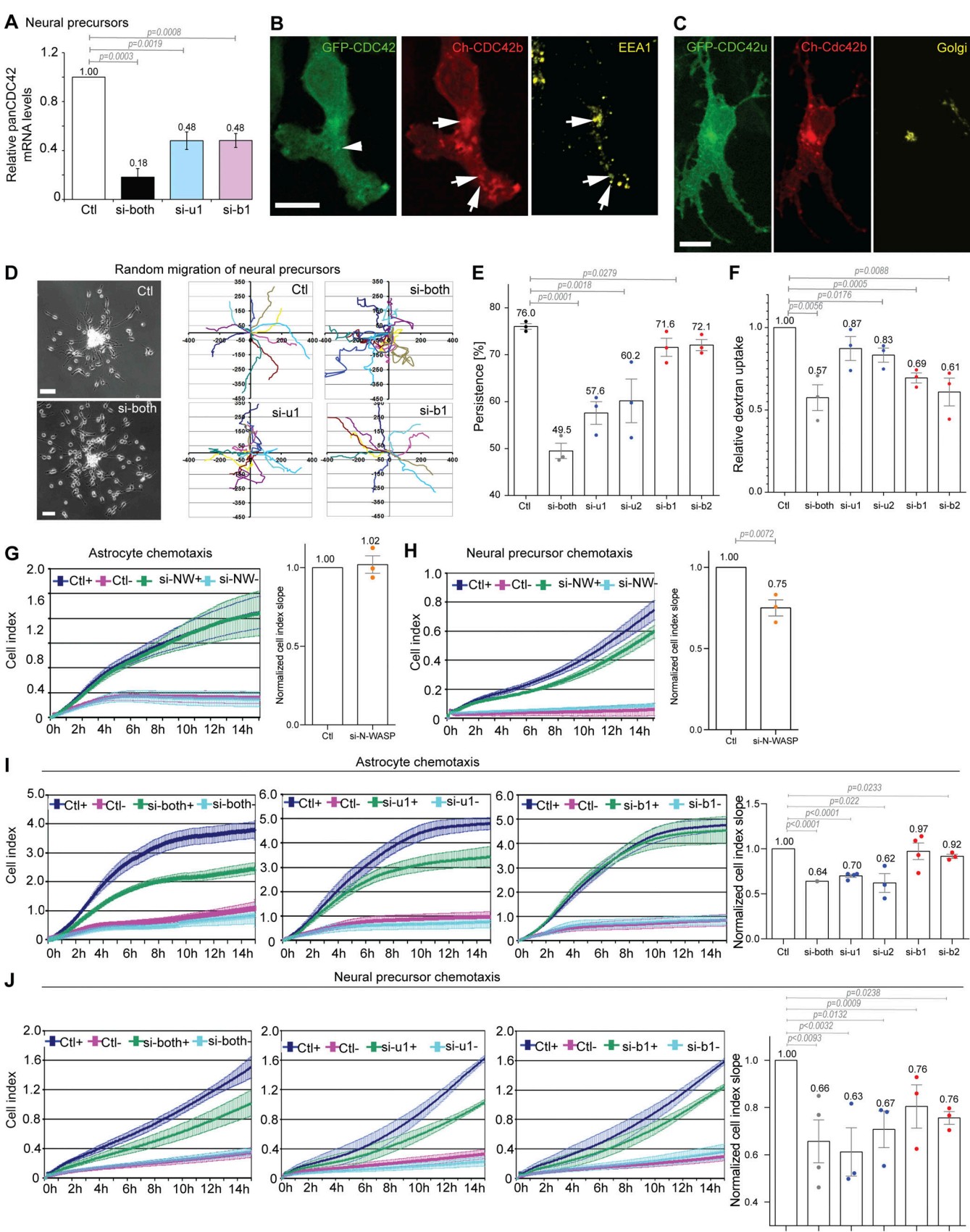

Figure 5. **CDC42 isoforms cooperate to promote neural precursor cell (NPC) chemotaxis. (A)** Total CDC42 mRNA levels were measured using a TaqMan assay that recognizes both CDC42 isoforms (panCDC42) in neural precursors transfected with the indicated siRNA. **(B and C)** Fluorescence images of NPCs

expressing GFP-CDC42u and mCherry-CDC42b, fixed and stained with anti EEA1 (early endosome marker) (B) and anti-GM130 (Golgi) (C). **(D)** Phase contrast images of NPCs nucleofected with the indicated siRNA and migrating out of neurospheres 5 h after plating. Right panels show the representative cell trajectories over 4 h of migration. **(E)** The directional persistence of NPC migration was measured between 100 and 300 min after plating. **(F)** Relative dextran uptake into NPCs transfected with the indicated siRNAs. The graph shows the data and the means ± SEM of three independent experiments with at least 150 cells analyzed per condition. **(G–J)** Astrocyte and NPC chemotaxis in Boyden chamber-based xCelligence system assays. +: bottom well contains FBS; −: no FBS in bottom well. Curves show the impedance measurement over time on the bottom surface of the filter. Right panels show the curve slopes, indicative of the rate of migration in chemotactic conditions in which FBS was contained in the bottom wells. Graphs show the data and the means ± SEM of at least three independent experiments. Data were normalized to the control. **(G and H)** Effects of N-WASP (si-NW) knockdown on chemotactic migration of astrocytes (G) or NPCs (H). **(I and J)** Chemotactic migration of astrocytes (I) or NPCs (J) upon knockdown of CDC42 isoforms. All P values were calculated using two-sided unpaired Student's *t* test. Scale bars: 100 µm.

---

respectively. RNA was isolated by using the RNeasy Mini Kit (QIAGEN) followed by the digestion of contaminating genomic DNA (Turbo DNA Free). RNA concentration and purity were determined by spectrophotometry. cDNA synthesis was performed according to kit instructions (VILO cDNA synthesis; Invitrogen). Quantitative real-time PCR was performed using Applied Biosystems custom TaqMan Gene Expression Assays designed using the Taqman Assay Search Tool (Life Technologies) for the ubiquitous CDC42 isoform. Assay Rn00821429_g1 was used to quantify panCDC42 mRNA levels (both isoforms). As an endogenous control, we used PGK1 (assay Rn00821429_g1) and Ppia (assay Rn00690933_m1) (all from Life Technologies) for astrocytes and Casc3 (Rn00595941_m1) and Eif2b1 (Rn00596951_m1) for neural precursor cells. Real-time PCR amplification was performed on a Sequence Detection System (7500; Applied Biosystems) using TaqMan Universal MMix II (Life Technologies) according to the manufacturer's instructions. Thermal cycling conditions were as follows: 50°C for 2 min, 95°C for 10 min followed by 40 cycles of 95°C for 15 s, and 60°C for 1 min. Data were collected and analyzed with the SDS software v2.0.6 (Applied Biosystems). The comparative CT method ($\Delta\Delta C_T$) was used as described in the AB7500 SDS guidelines.

### Live spinning disk confocal microscopy
To study protein localization in live cells, plasmids encoding the GFP- or mCherry-tagged protein were microinjected into wound border astrocytes plated on glass bottom dishes (MatTek) and scratched 1 h before microinjection. Live imaging was done 4–8 h later using a spinning disk confocal microscope (Perkin Elmer Ultra View ERS) equipped with a heating chamber (37°C) and $CO_2$ supply (5%).

### Centrosome/Golgi reorientation and PKCζ localization assays
Scratch-induced cell polarization of astrocytes was studied as detailed previously (Etienne-Manneville, 2006). Briefly, a monolayer of astrocytes plated on poly-*L*-ornithine (Sigma-Aldrich)-coated coverslips was scratched and fixed 8 h later followed by immunostaining of centrosome, Golgi, and microtubules. Wound border cells were counted as polarized when the centrosome and Golgi were situated within the 90°C or 120°C section of a virtual circle drawn around the nucleus that faces the wound (see Fig. 4 B). Images were acquired on a Leica DM6000 epifluorescence microscope equipped with 40×, NA 1.25 and a 63×, NA 1.4 objective lenses and were recorded with a CCD camera (CoolSNAP HQ, Roper Scientific) using Leica LAS

AF software. PKCζ accumulation at the cell front was studied in knockdown cells situated on poly-L-ornithine coated coverslips and fixed 4 h after wounding.

### Cell spreading assays
Sixteen-well E-plates (Ozyme) were precoated with 25 µg/ml Fibronectin (Sigma-Aldrich) for NPCs or used uncoated for astrocytes. Single-cell suspensions of $1 \times 10^5$ NPC or $5 \times 10^4$ astrocytes in 100 µl of normal culture medium were added per well. Each condition (control or siRNA transfected) was run in triplicate wells. Cell adhesion and spreading were measured as changes in impedance measured on the xCelligence RTCA DP Analyzer (ACEA Biosciences) according to the manufacturer's instructions.

### Cell migration assays
Astrocyte wound healing and neurosphere migration assays: For live imaging of astrocyte wound healing experiments, transfected astrocytes were seeded into poly-*L*-ornithine (Sigma-Aldrich)-coated 12-well standard plastic dishes using a normal cell culture medium. Cell monolayers were scratched immediately before image acquisition followed by the addition of 20 mM HEPES (Sigma-Aldrich) as well as an antioxidant (Sigma-Aldrich) to the medium and addition of liquid paraffin on top of the medium. Video time-lapse data were acquired on a Zeiss Axiovert 200 M microscope equipped with a 37°C humidified heating chamber as well as $CO_2$ supply (5%) by taking pictures every 15 min over 16 h and analyzed by manual tracking of cells using ImageJ, as described previously (Camand et al., 2012). For tracking of NPCs, neurospheres obtained from suspension cultures of transfected cells were plated into fibronectin-coated 12-well plastic dishes immediately before live imaging followed by the addition of HEPES, antioxidant and paraffin as for astrocytes. Images were taken every 5 min for NPCs.

Chemotaxis assays: The bottom wells of 16-well C.I.M plates (Ozyme) were filled with 160 µl of normal culture medium for astrocytes or NPCs containing 10% FBS in both cases. The upper wells were filled with 50 µl of normal culture medium without FBS (but containing B27 in case of NPCs). After 1 h of equilibration in the incubator, single-cell suspensions of $1 \times 10^5$ NPC or $5 \times 10^4$ astrocytes in 100 µl medium (without FBS) were added to the upper wells plated in the upper wells. Each condition (control or siRNA transfected) was run in triplicate wells on the xCelligence RTCA DP Analyzer according to user guidelines. Cell migration was measured as changes in impedance with the slopes of cell migration being compared over 10 h.

## Endocytosis assays

Neurospheres were incubated for 1.5 h at 37°C in 1 mg/ml Texas Red labeled dextran (10 kD Invitrogen), and extensively washed and plated on poly-*L*-ornithine and fibronectin-coated glass coverslips for 4 h before being fixed in 4% paraformaldehyde. Transfected astrocytes were plated on poly-*L*-ornithine coated coverslips and scratched 4 d later, followed by the addition of 1 mg/ml fluorescent dextran. Cells were fixed in PFA after 1.5 h and analyzed on the Leica DM6000 epifluorescence microscope described above. The relative amount of dextran uptake was determined using ImageJ to quantify the number of fluorescence maxima per cell after background subtraction.

## Pulldown assays

Wild-type HEK cells were used to test interactions with an endogenous protein. Cells were lysed using pull-down buffer containing 500 mM NaCl, 15 mM KCl, 8 mM TRIS base, 12 mM HEPES free base, 3 mM $MgCl_2$, 10% (vol/vol) glycerol, 1% (vol/vol) Triton X-100 as well as 1 tablet/10 ml protease inhibitor Mini-complete, EDTA-free (Roche). After the removal of insoluble compounds via centrifugation, lysates were incubated with glutathione Sepharose beads (GE Healthcare) coated with the other interaction partner for 30 min at 4°C on a rotary wheel. Subsequently, beads were washed three times for 10 min with the same buffer containing however doubled amounts of NaCl on the rotary wheel, followed by the detection of associated proteins using Western blotting.

## Online supplemental material

Fig. S1 includes the study of CDC42 isoform expression levels in the cells used in this study. Fig. S2 presents a more complete analysis of the Proteomic screen of CDC42 effectors, GEFs, and GAPs. Fig. S3 shows the effect of palmitoylation and geranylgeranylation inhibitors on CDC42 localization and additional controls. Video 1 illustrates the distinct intracellular localization of CDC42u and CDC42b in immobile astrocytes. Video 2 shows GFP-CDC42u and Video 3 GFP-CDC42b in migrating astrocytes. Video 4 shows that mCherry-CDC42b colocalizes with GFP-N-WASP on endocytic vesicles.

## Data availability

The data are available from the corresponding author upon reasonable request.

## Acknowledgments

We would like to thank members of the SEM lab for support and discussion. We gratefully acknowledge the Imagopole of Institut Pasteur (Paris, France).

This work was supported by the Centre National de la Recherche Scientifique and the Institut Pasteur. Y. Ravichandran was funded by the Polarnet ITN (Innovative Training Network) part of the European Commission and the Fondation pour la Recherche Médicale. J. Hänisch was funded by a Marie Curie post-doctoral grant. B. Boëda is a full-time Institut National de la Santé et de la Recherche Médicale researcher.

Author contributions: Y. Ravichandran: Investigation, Methodology, Validation, Visualization, Writing—Original draft. J. Hänisch: Methodology, Investigation, Visualization, Writing—Original draft. K. Murray: Methodology, Investigation, Visualization. V. Roca: Methodology, Investigation. F. Dingli: Investigation, Visualization. D. Loew: Supervision, Data Curation. V. Sabatet: Data Curation, Visualization, Formal analysis. B. Boëda: Supervision, Conceptualization, Methodology, Visualization, Writing—Reviewing and Editing. T. Stradal: Conceptualization, Resources. S. Etienne-Manneville: Supervision, Conceptualization, Funding acquisition, Project administration, Writing—Original draft, Writing—Reviewing and Editing.

Disclosures: The authors declare no competing interests exist.

Submitted: 14 April 2020

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

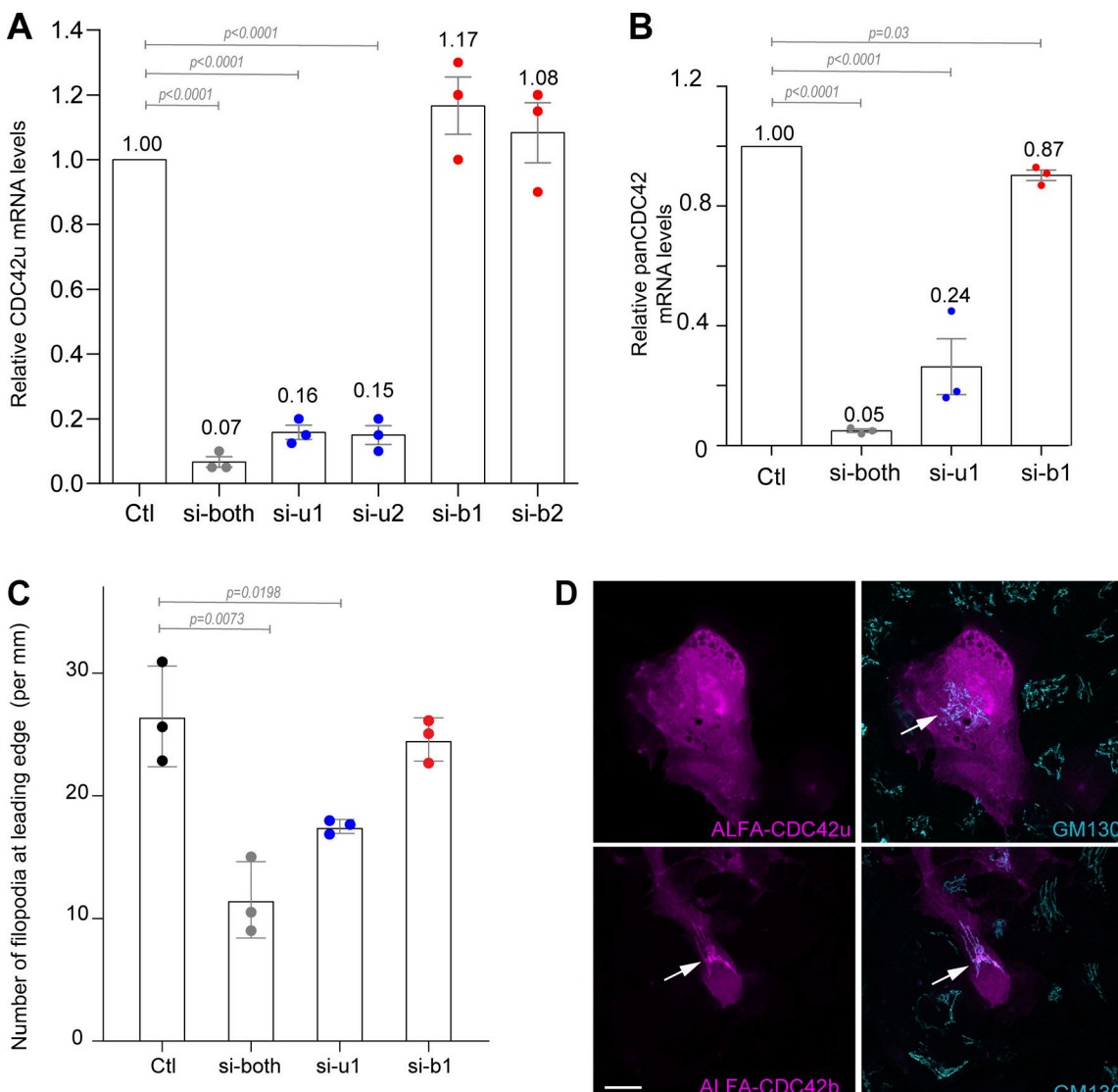

Figure S1. **Differential expression and localization of CDC42 isoforms in astrocytes. (A–D): (A)** QPCR data using a TaqMan assay specifically detecting the ubiquitous CDC42u in astrocytes transfected with the indicated siRNA specific for each distinct isoform (si-u1, si-u2 and si-b1, si-b2) or targeting both isoforms (si-both). **(B)** Total CDC42 mRNA levels were measured using a TaqMan assay that recognizes both CDC42 isoforms (panCDC42) in astrocytes transfected with the indicated siRNA. **(C)** Graph showing the number of filopodia formation per mm length of wound edge post 1 h of scratch wound in astrocytes transfected with the indicated siRNA. **(D)** Confocal images of fixed astrocytes expressing ALFA-tagged CDC42 constructs 4–6 h after microinjection and stained with cis-Golgi marker GM130.

**A**

| Effectors | | | | | | | |
|---|---|---|---|---|---|---|---|
| | Cdc42b-CA/GFP | | | | Cdc42u-CA/GFP | | |
| Proteins | Ratio | Log2 | p-value | Peptides used | Ratio | Log2 | p-value | Peptides used |
| MAP3K4 | 0.54 | -0.88 | 1.27E-03 | 36 | 0.55 | -0.86 | 1.01E-04 | 31 |
| MRCKγ | 0.87 | -0.21 | 7.99E-01 | 29 | 1.39 | 0.47 | 5.54E-01 | 33 |
| IQGAP1 | 53.00 | 5.73 | 1.60E-188 | 430 | 53.74 | 5.75 | 3.14E-205 | 409 |
| MAP3K11 | 0.72 | -0.48 | 2.26E-01 | 8 | 0.75 | -0.41 | 2.44E-01 | 8 |
| PAR-6A | 1000.00 | 9.97 | | 27 | 1000.00 | 9.97 | | 26 |
| PAK6 | 1000.00 | 9.97 | | 8 | 1000.00 | 9.97 | | 6 |
| PKN1 | 0.84 | -0.25 | 2.56E-02 | 52 | 0.68 | -0.55 | 1.16E-04 | 55 |
| IQGAP2 | 37.33 | 5.22 | 1.39E-59 | 180 | 38.04 | 5.25 | 1.81E-68 | 169 |
| IQGAP3 | 25.53 | 4.67 | 8.25E-53 | 178 | 35.96 | 5.17 | 4.94E-59 | 170 |
| SPEC2 | 58.00 | 5.86 | 3.71E-08 | 16 | 55.95 | 5.81 | 5.29E-10 | 16 |
| BORG5 | 52.45 | 5.71 | 1.12E-51 | 89 | 50.70 | 5.66 | 9.19E-51 | 85 |
| PIK3R1 | 1.75 | 0.81 | | 5 | 0.99 | -0.01 | | 5 |
| PAR-6G | 8.42 | 3.07 | 2.16E-05 | 16 | 9.75 | 3.28 | 2.82E-05 | 15 |
| DAAM1 | 4.59 | 2.20 | 3.60E-07 | 38 | 4.72 | 2.24 | 2.21E-07 | 37 |
| SPEC1 | 17.01 | 4.09 | 7.61E-04 | 15 | 8.68 | 3.12 | 1.93E-02 | 14 |
| ACK1 | 6.53 | 2.71 | 3.42E-07 | 25 | 6.20 | 2.63 | 7.87E-06 | 20 |
| BORG2 | 0.57 | -0.80 | 6.15E-01 | 7 | 0.78 | -0.35 | 8.14E-01 | 7 |
| PAK1 | 2.73 | 1.45 | 4.99E-03 | 15 | 2.41 | 1.27 | 3.00E-02 | 14 |
| PAK2 | 11.53 | 3.53 | 4.93E-13 | 49 | 10.22 | 3.35 | 9.03E-13 | 48 |
| FMNL2 | 4.46 | 2.16 | 2.37E-04 | 15 | 7.68 | 2.94 | 9.27E-06 | 16 |
| MRCKβ | 15.90 | 3.99 | 1.75E-19 | 101 | 27.12 | 4.76 | 7.28E-25 | 99 |
| CIP4 | 4.37 | 2.13 | 4.16E-01 | 8 | 5.49 | 2.46 | 2.49E-01 | 10 |
| IRSp53 | 12.60 | 3.65 | 3.55E-09 | 33 | 5.99 | 2.58 | 9.63E-07 | 31 |
| MRCKα | 13.11 | 3.71 | 7.51E-60 | 212 | 13.50 | 3.75 | 1.01E-52 | 208 |
| N-WASP | 25.36 | 4.66 | 1.19E-21 | 58 | 44.89 | 5.49 | 1.49E-25 | 54 |
| PAR-6B | 13.30 | 3.73 | 1.93E-17 | 56 | 16.47 | 4.04 | 5.87E-19 | 53 |
| BORG1 | 64.10 | 6.00 | 3.06E-20 | 41 | 70.40 | 6.14 | 5.99E-16 | 35 |
| PAK4 | 36.67 | 5.20 | 9.07E-38 | 64 | 52.29 | 5.71 | 1.38E-42 | 59 |
| BORG4 | 188.52 | 7.56 | 1.04E-11 | 22 | 242.06 | 7.92 | 1.85E-12 | 22 |

**B**

| GEFs | | | | | | | |
|---|---|---|---|---|---|---|---|
| | Cdc42b-CA/GFP | | | | Cdc42u-CA/GFP | | |
| Protein | Ratio | Log2 | p-value | Peptides used | Ratio | Log2 | p-value | Peptides used |
| α-PIX | 1.23 | 0.29 | 8.59E-01 | 8 | 2.26 | 1.18 | 2.66E-02 | 9 |
| RhoGEF | 0.54 | -0.88 | 1.21E-01 | 8 | 0.41 | -1.27 | 7.52E-03 | 12 |
| DOCK7 | 0.41 | -1.29 | 2.27E-32 | 184 | 0.50 | -1.01 | 2.78E-25 | 179 |
| DOCK6 | 0.96 | -0.06 | 9.11E-01 | 14 | 1.31 | 0.39 | 2.37E-01 | 13 |
| DOCK9 | 1.02 | 0.02 | 9.59E-01 | 13 | 1.07 | 0.09 | 8.84E-01 | 11 |
| β-PIX | 2.03 | 1.02 | 2.46E-02 | 17 | 1.96 | 0.97 | 4.14E-01 | 11 |
| Tuba | 7.89 | 2.98 | 8.00E-22 | 119 | 9.44 | 3.24 | 2.74E-22 | 117 |

**C**

| GAPs | | | | | | | |
|---|---|---|---|---|---|---|---|
| | Cdc42b-CA/GFP | | | | Cdc42u-CA/GFP | | |
| Proteins | Ratio | Log2 | p-value | Peptides used | Ratio | Log2 | p-value | Peptides used |
| RICH1 | 2.07 | 1.05 | 5.84E-04 | 29 | 1.00 | 0.01 | 9.90E-01 | 27 |
| DYNLL1 | 0.22 | -2.22 | 1.28E-05 | 16 | 0.22 | -2.19 | 3.39E-06 | 16 |
| RACGAP1 | 0.70 | -0.52 | 7.51E-03 | 28 | 0.64 | -0.64 | 3.12E-03 | 30 |
| STARD13 | 1.04 | 0.06 | 9.50E-01 | 8 | 1.04 | 0.06 | 8.24E-01 | 9 |
| DLC1 | 8.86 | 3.15 | 3.07E-03 | 8 | 7.85 | 2.97 | 1.02E-03 | 8 |
| RhoGAP31 | 1000.00 | 9.97 | | 3 | 1000.00 | 9.97 | | 4 |

Figure S2. **Proteomic screen of CDC42 effectors, GEFs, and GAPs. (A)** The table represents the list of known CDC42 interactors found in our screen in comparison to the list of known interactors in the field listed in Pichaud et al. (2019). **(A–C)** The screen was divided into three categories: CDC42 effectors (A), CDC42 GEFs (B), and CDC42 GAPs (C). Columns ratio refers to the number of peptides bound to respective CDC42 isoform normalized to GFP. Log$_2$ values were calculated for these ratios for ease of representation on correlation plots in Fig. 2, E–G. P values and peptides used for the calculation are represented. See Materials and methods for proteomic screen details and PRIDE database for screen raw data.

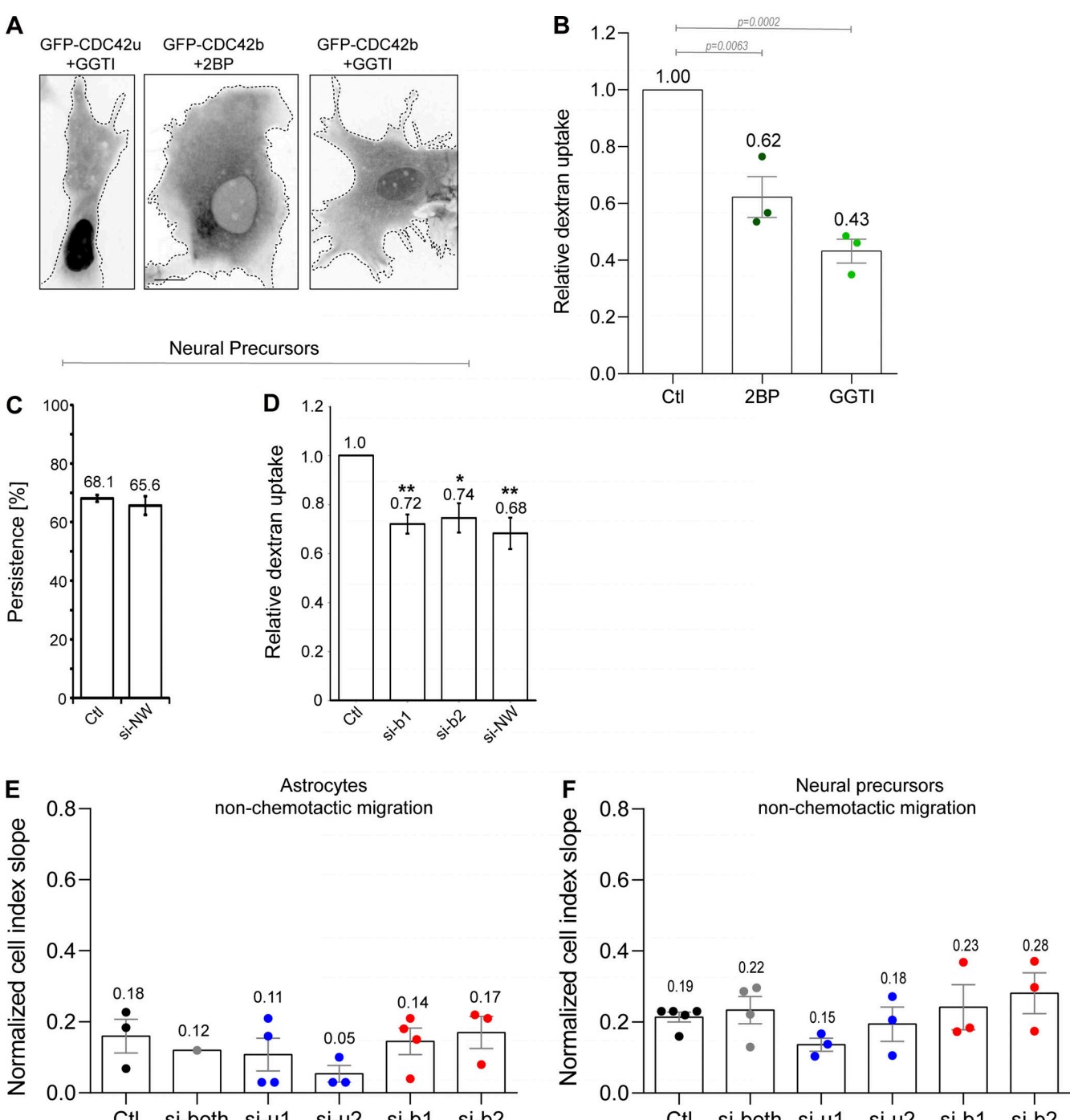

Figure S3.  **Role of CDC42 isoforms in dextran uptake and in non-chemotactic migration. (A)** Localization of GFP-CDC42b or GFP-CDC42u in astrocytes pretreated overnight with an inhibitor of palmitoylation (2-bromopalmitate, 2BP, 120 μM) or an inhibitor of geranylgeranylation and/or palmitoylation (GGTI, 20 μM). **(B)** Histogram showing the relative dextran uptake observed in treated cells compared to control cells (Ctl). The graph shows the data and means ± SEM from three independent experiments. At least 150 cells per condition were analyzed. All P values were calculated using a two-sided unpaired Student's *t* test. Scale bars: 10 μm. **(C)** Directional persistence (measured between 100 and 300 min after plating) of NPCs transfected with the indicated siRNA and migrating out of neurospheres. **(D)** Quantification of relative dextran uptake in NPCs transfected with the indicated siRNA. **(E and F)** Astrocyte (E) and NPC (F) migration in Boyden chamber-based xCelligence system assays. Graphs show the curve slopes indicative of the rate of migration in non-chemotactic conditions (no FBS in the lower well). Histograms show data points and means ± SEM of three to five independent experiments. At least 250 cells per condition were analyzed. All P values were calculated using two-sided unpaired Student's *t* test. Scale bars: 10 μm.

Video 1. **CDC42u and CDC42b display distinct intracellular localization.** Video showing mCherry-CDC42u and GFP-CDC42b expressing astrocytes. Fluorescent images were taken using a spinning disk microscope every 10 s (total length: 12:20 min). Bar: 10 μm.

Video 2. **GFP-CDC42u is mainly cytosolic.** Video showing a GFP-CDC42u expressing migrating astrocyte, 5 h after wounding. Fluorescent images were taken using a spinning disk microscope every 10 s (total length: 13:20 min). Bar: 10 μm.

Video 3. **GFP-CDC42b localizes to Golgi, vesicles, and the plasma membrane.** Video showing a GFP-CDC42b expressing migrating astrocyte 5 h after wounding. Fluorescent images were taken using a spinning disk microscope every 10 s (total length: 20 min). Bar: 10 μm.

Video 4. **mCherry-CDC42b colocalizes with GFP-N-WASP on endocytic vesicles.** Video showing a migrating astrocyte expressing mCherry-CDC42b and GFP-N-WASP 5 h after wounding. Fluorescent images were taken using spinning disk microscopy every 10 s (total length: 20 min). Bar: 10 μm.

