## [Peer Review File · The Journal of Cell Biology]

The distinct localization of CDC42 isoforms is responsible for specific functions during migration

Yamini Ravichandran, Jan Hänisch, Kerren Murray, Vanessa Roca, Florent Dingli, Damarys Loew, Valentin Sabatet, Batiste Boëda, Theresia Stradal, and Sandrine Etienne-Manneville

Corresponding Author(s): Sandrine Etienne-Manneville, Institut Pasteur

Review Timeline:

Submission Date:	2020-04-14
Editorial Decision:	2020-05-20
Revision Received:	2023-07-26
Editorial Decision:	2023-08-25
Revision Received:	2023-09-20

Monitoring Editor: Ian Macara

Scientific Editor: Andrea Marat

Transaction Report:

DOI: <https://doi.org/10.1083/jcb.202004092>

May 20, 2020

Re: JCB manuscript #202004092

Dr. Sandrine Etienne-Manneville
Institut Pasteur
Institut Pasteur
25 rue du Dr Roux
Paris 75015
France

Dear Sandrine,

I had been asked to serve as the editor for your interesting manuscript on "The specific localization and functions of Cdc42 isoforms". Your work has now been reviewed by three outside referees with expertise in this field, and their comments are appended. As you will see, they all appreciate the value in defining the differences in biological function between the two Cdc42 variants, but they raise several important questions about the validity of the conclusions, as outlined below. Unfortunately, for these reasons we cannot accept the manuscript for publication in its present form. Responding to the reviewer concerns will require additional experimental work. I don't know how easy this is in the current circumstances (our laboratories are beginning to partially re-open, but I think some universities in the US are still closed until the end of May, and I don't know the current situation in France). One alternative would be to transfer the manuscript to another journal such as the LSA, which would not require further experimental work.

We have discussed your manuscript with the editors of Life Science Alliance (<http://www.life-science-alliance.org/>) and they would like to offer to publish your study provided you provide a point-by-point response that diligently addresses the concerns of the reviewers and according changes to the manuscript text and data representation. LSA is our academic editor-led, open access journal launched as a collaboration between RUP, EMBO Press and Cold Spring Harbor Press. You can use the link below to initiate an immediate transfer of your manuscript files and reviewer comments to LSA.

Link Not Available

If you wish to revise for JCB :

The key points from your study are that (1) there is no difference in the interactions of the 2 Cdc42 variants with effector proteins; (2) the two variants have different subcellular localizations; and (3) the brain variant primarily regulates endocytosis while the placental variant regulates polarity functions involved in cell migration.

Reviewer #1 is concerned about point (1). They note that an explanation is required of the metric used to determine that there is no effector binding difference; and notes that "In Fig S2B, there are clear differences between some effectors (eg CDC42SE1, Log2 4.1 for BR-CA versus Log2 3.1 for PL-CA). How is this not differential interaction?" This reviewer also notes that the proteomics only shows what CAN interact, not what DOES interact with the two variants. One would need to use BiID or some similar proximity labeling method to determine if there are differences in what DOES bind.

Reviewer #2 has concerns with conclusion (3). They point out that "The most critical gap is in connection with their conclusion that "brCdc42, embodies the major isoform regulating endocytosis". This is based only on analysis of macropinocytosis, a process in which Cdc42 is a secondary regulator. Yet they ignore the major Cdc42-dependent CLIC/GEEC pathway." Support for this conclusion would require analysis of clathrin dependent and independent, Cdc42-dependent endocytosis. They also wonder if the differences between variants could be because of differences in the level of activation to the GTP-bound states (however, I do not feel that this would be necessary to determine experimentally, for this study).

Reviewer #3, like reviewer #1, points out that there are in fact some differences in binding data for the two variant Cdc42s, noting that "There is one protein outlier that binds much more to plaCdc42 than BrCdc42". They also have questions about the localization data (conclusion (2)) based on the movies, and are concerned that the GFP tag might interfere with localization (note that Sophie Martin has found that GFP-Cdc42 is not fully functional in *S. pombe*).

The reviewers also ask that the recent Cerione paper be included and discussed; that data points be included for all figures; that the levels of Cdc42 depletion be presented; that metrics for localization be provided; that levels of GFP-Cdc42 relative to endogenous protein be provided; and that the naming of the variants be uniform and in accord with previous publications.

If you do decide to submit a revised manuscript, please note that we will need a point-by-point response to each of the reviewer comments, and that the manuscript will be returned to at least 2 of the original reviewers for their comments.

Sincerely,

Ian Macara, Ph.D.
Editor

Andrea L. Marat, Ph.D.
Scientific Editor

Journal of Cell Biology

Reviewer #1 (Comments to the Authors (Required)):

The manuscript by Hänisch et al. examine the cell biology of alternate transcripts of Cdc42. In contrast to the ubiquitous Cdc42 isoform (referred to as 'placental Cdc42'), the authors describe a 'brain isoform of Cdc42' that due to differential lipid modification of the c-terminus localises to both early endosomes and the Golgi complex. Surprisingly, the authors report no apparent difference in the interactome of these two Cdc42 variants. Instead, differential localisation of Cdc42 isoforms is proposed to regulate differential function. While 'placental' Cdc42 fulfils the major previously described roles for Cdc42 in astrocyte cell migration, 'brain' Cdc42 is required for pinocytic Cdc42 functions, and therefore chemotactic migration.

Despite wide-spread alternate splicing of transcripts, relatively little is known about how the resultant alternate isoforms shape the endomembrane system. This is therefore an interesting dive into the different functions of Cdc42. There are a number of control experiments and clarifications that could make a nice manuscript improve to the level of JCB.

Points to address

1. The authors report that, from proteomics of Cdc42 isoform interactors, that they "did not show any significant difference between both isoform." Presented are plots without significance indicated. In Fig S2B, significance values are indicated from what appears to be pulldown against GFP alone. Could significance be calculated between isoforms? Conversion of data (Fig 2B, C) into volcano plots would allow the reader to appreciate significance.

What is the metric used to determine that there is no effector binding difference between isoforms? Is the amount of CDC42 precipitated used as a normalizing ratio to determine interactor association? For instance, in Fig S2A there is a Log2 value of 6 difference in the amount of Cdc42 itself detected between the pulldowns of interactome experiments between isoforms. The value for Cdc42 is missing from Fig 2SB. How does the amount of Cdc42 detected influence the data? In Fig S2B, there are clear differences between some effectors (eg CDC42SE1, Log2 4.1 for BR-CA versus Log2 3.1 for PL-CA). How is this not differential interaction?

2. The proposed concept of 'no-difference-between-Cdc42-isoform-interactomes' is difficult to reconcile with the Cdc42 isoform 'differential-function-based-on-localisation' notion reported. This suggests that all Cdc42 interactors are at all locations in the cell where either Cdc42 isoform is localised. How then are alternate Cdc42 effects regulated? The 'same' protein complex, just in a different location?

This may partially be explained by the methods utilised. With an emphasis on differential localisation of Cdc42 isoforms, it is curious that the authors use GFP-trap affinity pulldown coupled with mass spectrometry to identify interactors. Along with the use of 'constitutively active'/hydrolysis-retarded mutations ('CA') of Cdc42, this will identify what CAN interact with these isoforms when localisation is destroyed by homogenisation and incubation in a tube. What is not shown is what DOES interact differently based on differential localisation. In this sense, proximity-ligation approaches would possibly give a clearer picture of the differential interactomes of Cdc42 variants in situ.

In the absence of providing such a clearer map of interactions, the authors need to be explicitly clear on this point of that their data identifies what CAN be the interactome, versus what physiologically is the interactome of alternate variants. This is not necessarily a caveat to the studies, but without overt clarification is it problematic as presented.

3. The nomenclature that the authors use for alternate isoforms of Cdc42 is problematic. The authors refer to:
Authors: brCdc42 (brain); Endo et al, JBC, 2020: Cdc42b (brain), Uniprot: Cdc42 Isoform 1/brain.
Authors: plaCdc42 (placental); Endo et al, JBC, 2020: Cdc42u (ubiquitous); Uniprot: Cdc42 Isoform 2/placental.

Since Cdc42 is not expressed in only brain and placenta, could the authors kindly be requested to unify the nomenclature to Isoform 1 and Isoform 2? This might make this easier for the field going forward.

4. Introduction: 'Lacunae' is an obscure word in English.

5. Statistical considerations. Quantifications of localisations are not consistent across the manuscript. In Fig 1C. How is the metric 'cells with strong Golgi enrichment' determined? What determines 'strong'? While phenotypes in Fig 1C, D, S1A are quantified, those in 1E are not. All phenotypes should be quantified, including immunoprecipitation experiments (Fig. S2C-E).

Individual Data points should be overlaid on graphs to allow the reader to understand the variability in experiments, particularly as the use of SEM poorly represents this. Three independent experiments are noted, but what are the sample sizes within experiments?

6. For Fig 1F, the authors state, "qPCR quantification of Cdc42 upon knockdown of each isoform revealed that roughly 15% of the total Cdc42 mRNA pool in astrocytes encodes the brain isoform (Fig. S1F) (Yap et al., 2016)." Should this be 13%?

7. The discussion section is very brief, and doesn't put the findings into the context of the field. Some mention of the above points, and why the brain isoform is required for Neural Precursor Cell, but not astrocyte, chemotaxis should be included.

8. Conspicuous by absence is mention of a recent paper by Richard Cerione's lab (PMID: 32071086; in JBC online in mid-February) studying the role of Cdc42 isoforms in neurogenesis. This paper, and how the mechanisms reported reflect on this manuscript's theory of localisation-dependent alternate isoform function, should at least be discussed.

9. Movies 5 and 6 are referred to in the text, but these are not in the files available or listed in the figure legends.

Reviewer #2 (Comments to the Authors (Required)):

This manuscript reports functional differences between two isoforms of Cdc42, the widely expressed prenylated placental form and the more restricted prenylated and palmitoylated brain isoform. They show that the PlaCdc42 localizes to the cytosol and plasma membrane, and controls polarized cell migration. BrCdc42 on the other hand localizes mainly to intracellular membranes and controls macropinocytosis, probably via N-Wasp. No difference could be detected in their interactions with effector proteins, suggesting that localization to different compartments is the determinant of differential function.

The experiments are generally done well and presented clearly, with a few minor weaknesses, noted below. There are a number of interesting new results but overall it is rather thin with notable gaps that limit its significance. They do not even look at filopodia. The most critical gap is in connection with their conclusion that "brCdc42, embodies the major isoform regulating endocytosis" (from the Abstract). This is based only on analysis of macropinocytosis, a process in which Cdc42 is a secondary regulator. Yet they ignore the major Cdc42-dependent CLIC/GEEC pathway. A more careful examination of clathrin and clathrin-independent, Cdc42-dependent (CLIC) endocytosis is required to support this main conclusion.

A second gap is whether the functional difference between isoforms is due to differential activation in the different settings. Is the brain isoform poorly activated in the wounding assay, and conversely, is the placental isoform poorly activated in the endocytosis assay?

Other issues:

Figure 2. what were the expression levels of the transfected GFP-Cdc42 constructs relative to endogenous and each other?
Fig 3E and 4B. What were the expression levels of the rescue constructs relative to endogenous and each other?

Reviewer #3 (Comments to the Authors (Required)):

This manuscript for the first time shows different roles of the two Cdc42 splice variants in cell migratory polarity versus pinocytosis. It provides a clear and thorough analysis, with appropriate controls. The assays are standard and not innovative, having been established many years ago. Although it is well established that Cdc42 is important in both assays, the Cdc42 splice variants have never been compared, and in most cases researchers are not even aware that there are two splice variants. This is therefore an important comparison.

1. The naming of the two Cdc42 isoforms is confusing, especially because plaCdc42 is not specific to the placenta, and the authors variably use capitals and small letters and sometimes use PICdc42. In addition, splice variant information is always placed after the gene name (e.g. Rac1b) rather than before (as done here). The Cerione group have just published in JBC using the nomenclature Cdc42u (ubiquitous) and Cdc42b (brain). It would be ideal if the same nomenclature is used here, consistent with them, rather than introducing a different nomenclature. The Cerione group paper should also be added to the Discussion here because it is highly relevant, although the analysis carried out is very different to that described here and not overlapping.
2. For brCdc42 (Fig. 1A), is it known if the CAAX box is not processed by removal of AAX followed by methylation? If it were processed, then palmitoylation would not occur. Is removal of the AAX regulatable?

3. Movies: for plaCdc42 there appears to be a lot of dynamic 'vesicular' movement, and it is not uniformly 'cytoplasmic' as the authors state in the text (movies 1 and 3). Some of it appears to overlap with brCdc42 (Movie 1). The text should describe exactly what is observed in the movies, unless the movie is unrepresentative of what they have found in multiple movies (and therefore should be replaced). By contrast brCdc42 'vesicles' are surprising static (movies 1 and 2) for the most part - this might be because overexpression impairs vesicle movement. There is also a concern that the fusion with GFP alters their ability to signal/localize correctly. What is the localization with a smaller, different tag?
4. Fig. 1C/D: how many cells were quantified in total and in each of the individual experiments? What is the definition of 'strong co-localization' with GM130/EEA1? Pearson's correlation coefficient should be used for both GM130 and EEA1 graphs to be consistent. What is the correlation coefficient for plaCdc42 and brCdc42? There are some fiber-like structures running across the images in Fig. 1C/D where they both appear to co-localize. What are these and are they observed in other images?
5. Fig. 1E: Why does brCdc42 without geranylgeranylation not go to the nucleus? What happens if both cysteines are mutated - does it then go to the nucleus? Can it be palmitoylated without prenylation (like RhoU/V)?
6. Fig. 2B: There is one protein outlier that binds much more to plaCdc42 than BrCdc42. Which protein is this - if it is Cdc42, then why was it not detected for brCdc42? It should be described in the results/labeled on the figure. Network analysis of the proteins identified would really enhance these important data - albeit not so relevant to the differences between the isoforms, the information of Cdc42-interacting proteins/complexes is still useful.
7. Fig. S1E - it would be useful to show the knockdown of brCdc42 on the mRNA level for comparison with plaCdc42, so that the relative knockdown of each isoform can be directly compared.
8. Fig. 3: For the graphs, all the data should be provided on the plot i.e. show a dot for each experiment, and also add graphs to show the range of data for all the cells tracked/analysed (dot plots). The total number of cells tracked or analysed should be given in the figure legend. Finally, absolute p values should be shown above each bar not asterisks (including those with no asterisk) - this is much preferred now. Please substitute the Greek letter for 'zeta' in the figure legends and figure.
9. Fig. 4: as for Fig. 3, give absolute p values and total numbers of cells analysed, adding graphs to show the variability in dextran uptake across all the cells analysed.
10. Figure S2: given that WASL is N-WASP, it is important to state this in the final column of the table, where it would be more helpful to give the abbreviated name rather than writing out the full name. This is also relevant for some other targets e.g. PAR6. Please use Greek letters in this column where relevant.
11. Fig. 5 and Fig S3: Amend graphs and figure legend information as for Fig. 3.
12. There are a few typos.

Point-by-point answers to the reviewers' comments

We thank the three reviewers for their constructive and encouraging comments. Please see our point-by-point responses (in blue) to their specific comments below. To facilitate the revision process we have also highlighted the newly added text in magenta in the submitted revised manuscript.

Referee:1

Points to address

1. The authors report that, from proteomics of Cdc42 isoform interactors, that they "did not show any significant difference between both isoform." Presented are plots without significance indicated. In Fig S2B, significance values are indicated from what appears to be pull-down against GFP alone. Could significance be calculated between isoforms? Conversion of data (Fig 2B, C) into volcano plots would allow the reader to appreciate significance. Cdc42 is a polarity protein that has several interactors even after strict screening of

What is the metric used to determine that there is no effector binding difference between isoforms? Is the amount of CDC42 precipitated used as a normalizing ratio to determine interactor association? For instance, in Fig S2 An there is a Log2 value of 6 differences in the amount of Cdc42 itself detected between the pull-downs of interactome experiments between isoforms. The value for Cdc42 is missing from Fig 2SB. How does the amount of Cdc42 detected influence the data? In Fig S2B, there are clear differences between some effectors (e.g. CDC42SE1, Log2 4.1 for BR-CA versus Log2 3.1 for PL-CA). How is this not differential interaction?

- We have modified the text to explain more clearly that our interpretation of the mass spec screen is qualitative in nature. The reviewer is correct, the normalization of the GFP-CDC42 proteins pulled down has been done with the GFP control to detect interactors specific to CDC42. It is not possible to obtain a strictly quantitative comparison of the binding partners.

Indeed, we cannot normalize the data by the amount of precipitated CDC42 for either isoform. As the reviewer correctly points out, despite several repeats, we always obtained much more CDC42u than CDC42b peptides in the analysis, although the expression of the two GFP-tagged isoforms is similar after transfection in HEK cells (see Figure A below). The reasons of this difference is not clear but may result from a different cleavage of the Cter.

Figure A: Western blot anti-CDC42 showing the level of expression of the transfected GFP-tagged proteins in total HEK cell lysates.

Normalization would lead us to conclude that CDC42b binds better to all interactors than CDC42u, which would contradict the more carefully controlled co-immunoprecipitation experiments. Moreover, the tagged proteins are both in large

excess compared to the endogenous interactors and that normalizing by the amount of CDC42 peptides does not have much sense as they are unlikely to be strictly limiting.

In this revised manuscript, we have improved our analysis by comparing our proteomic screen with the CRAPome database. In the CRAPome, negative control experiments generated by research groups across the world are collected (raw mass spectrometry data), reprocessed using a common pipeline, and annotated. This allowed us to refine the list of CDC42 interactors and compare it to the list of published interactors. Our interactome revealed 29 out of the 40 known epithelial effector proteins of CDC42 (Pichaud et al, J Cell Sci, 2019). We could detect 7 known GEFs and 5 known GAPs. All of these proteins were found associated both with CDC42u and CDC42b in the GFP-trap pull-down (Fig. 2D-F).

Thus, we have rephrased our conclusions saying that both isoforms can bind to the same effectors, as they are capable of pulling down similar proteins. For representative purposes and in order to highlight the most interesting interactors (known effectors, GEFs and GAPs identified in or pull-down) we have used correlation plots to compare the proteins associated with each isoforms. However, the requested complete volcano plots can be found in the PRIDE {Vizcaino et al., Nucleic Acids Res, 2016} partner repository with the dataset identifier PXD017477 (username: reviewer51683@ebi.ac.uk, Password: 4fNr03LX).

Finally, we would like to point out that a more quantitative comparison of the interactions directly relevant to this study (CDC42 isoforms functions in cell migration), we have performed GFP-immunoprecipitation assays and confirmed that there are no significant difference between CDC42 isoforms in their ability to bind N-WASP, PAR6 and aPKC ζ .

2. The proposed concept of 'no-difference-between-Cdc42-isoform-interactomes' is difficult to reconcile with the Cdc42 isoform 'differential-function-based-on-localisation' notion reported. This suggests that all Cdc42 interactors are at all locations in the cell where either Cdc42 isoform is localised. How then are alternate Cdc42 effects regulated? The 'same' protein complex, just in a different location? This may partially be explained by the methods utilised. With an emphasis on differential localisation of Cdc42 isoforms, it is curious that the authors use GFP-trap affinity pulldown coupled with mass spectrometry to identify interactors.

- In pulldown experiments the overexpressed tagged CDC42 isoforms show similar localization in the cytosol, on intracellular membrane including Golgi apparatus, at the plasma membrane (Shown in revised figure 2G). In these conditions, both isoforms can interact with various binding partners either in the cell or after cell lysis. This leads us to conclude the 'no-difference-between-CDC42-isoform-interactomes'.

While this biochemical analysis shows that, in principle, the two isoforms can similarly bind to the same effectors, but does not mean that the endogenous proteins which display distinct subcellular localization interact with the same effectors in cell. Low level of expression of the isoforms, using microinjection, shows that they localize differently. This was confirmed using a GFP, mCherry and a smaller Alfa tag (Revised Fig. S1F). Their interactions then also depend on the localization of the interactors, which is likely to depend on other factors (phosphoinositides, other proteins etc....). We now also show that CDC42b but not CDC42u co-localize with N-WASP on intracellular vesicles (revised figure 4F).

- We have clarified this point in the results and the discussion sections of the revised manuscript.

Along with the use of 'constitutively active'/hydrolysis-retarded mutations ('CA') of Cdc42, this will identify what CAN interact with these isoforms when localisation is destroyed by homogenisation and incubation in a tube. What is not shown is what DOES interact differently based on differential localisation. In this sense, proximity-ligation approaches would possibly give a clearer picture of the differential interactomes of Cdc42 variants in situ.

In the absence of providing such a clearer map of interactions, the authors need to be explicitly clear on this point of that their data identifies what CAN be the interactome, versus what physiologically is the interactome of alternate variants. This is not necessarily a caveat to the studies, but without overt clarification is it problematic as presented.

- We totally agree with the reviewer that overexpressed proteins CAN interact similarly with the same effectors but that endogenous proteins do not necessarily DO in cells. Unfortunately, all biochemical approaches such as BioID, require the overexpression of the precipitated proteins, and thus, will certainly not be able to show any difference in protein interactors. The proximity ligation assay would be very interesting if performed with antibodies detecting the endogenous proteins, which we have not been able to find or to produce. As it is, we instead, looked at the colocalization of fluorescently tagged CDC42 isoforms and N-WASP (revised Fig 4F).

3. The nomenclature that the authors use for alternate isoforms of Cdc42 is problematic. The authors refer to:

Authors: brCdc42 (brain); Endo et al, JBC, 2020: Cdc42b (brain), Uniprot: Cdc42 Isoform 1/brain.

Authors: plaCdc42 (placental); Endo et al, JBC, 2020: Cdc42u (ubiquitous); Uniprot: Cdc42 Isoform 2/placental.

Since Cdc42 is not expressed in only brain and placenta, could the authors kindly be requested to unify the nomenclature to Isoform 1 and Isoform 2? This might make this easier for the field going forward.

- We agree with the reviewers with regard to nomenclature. Following the reviewer recommendation and the nomenclature used by Endo et al, JBC, 2020, we have changed the text and the figures as follows:
 - Brain Cdc42 (brCdc42) to CDC42b
 - Placental Cdc42 (plaCdc42) to CDC42u (ubiquitous).

4. Introduction: 'Lacunae' is an obscure word in English.

- We replaced 'Lacunae' by "limitations"

5. Statistical considerations. Quantifications of localisations are not consistent across the manuscript. In Fig 1C. How is the metric 'cells with strong Golgi enrichment' determined? What determines 'strong'?

- For Fig.1C (now revised Fig 3B) Pearson correlation coefficient with JACoP plugin from ImageJ has been calculated to quantitatively assess the colocalization of Cis-Golgi-marker GM130 and CDC42 isoforms. A similar quantitative analysis has also been performed on CDC42 colocalization with EEA1 (revised Fig 3C).

While phenotypes in Fig 1C, D, S1A are quantified, those in 1E are not. All phenotypes should be quantified, including immunoprecipitation experiments (Fig. S2C-E).

- The co-immunoprecipitations shown in Fig.S2C-E (now revised Fig.2B and Fig.4H) have been quantified. The associated graphs show the data points of 3 independent experiments and the resulting mean and SEM.

Individual Data points should be overlaid on graphs to allow the reader to understand the variability in experiments, particularly as the use of SEM poorly represents this. Three independent experiments are noted, but what are the sample sizes within experiments?

- The data point of each independent experiments now given in all graphs, following the recommendations of J Cell Biol (Lord et al. J Cell Biol, 2020) Exact p-values (where $p < 0.05$ correspond to statistical significance) have also been included in all graphs.

6. For Fig 1F, the authors state, "qPCR quantification of Cdc42 upon knockdown of each isoform revealed that roughly 15% of the total Cdc42 mRNA pool in astrocytes encodes the brain isoform (Fig. S1F) (Yap et al., 2016)." Should this be 13%?"

- Figure S1D (previous S1F) shows that the siRNA directed against CDC42b decreases the total amount of CDC42 mRNA by 13%. We think it is unlikely that the siRNA approach totally abolish the expression of the CDC42b, so 13% would be an underestimation of the proportion of CDC42b. Moreover, we also noticed that siRNA-mediated downregulation of CDC42b tends to increase CDC42u mRNA level by approximately 10% (revised Fig. S1E). On the other hand, the siRNA directed against CDC42u leads to a decrease of 76% of the total CDC42 mRNA, which means that CDC42u represents at least 76% of the total CDC42 (and therefore CDC42b represents a maximum of 24%).

For these reasons, it is difficult to give the exact proportion of CDC42b and CDC42u and we have now written that "roughly 15-20% of total CDC42 mRNA" encodes the brain isoform (text p5).

7. The discussion section is very brief, and doesn't put the findings into the context of the field. Some mention of the above points, and why the brain isoform is required for Neural Precursor Cell, but not astrocyte, chemotaxis should be included.

- The discussion has been modified to answer the reviewer comment. P10, 11.

8. Conspicuous by absence is mention of a recent paper by Richard Cerione's lab (PMID: 32071086; in JBC online in mid-February) studying the role of Cdc42 isoforms in neurogenesis. This paper, and how the mechanisms reported reflect on this manuscript's theory of localisation-dependent alternate isoform function, should at least be discussed.

- We sincerely apologize for not having mentioned this paper, which came out just before the initial submission of our manuscript. The Cerione's group paper (Endo et al., J Cell Biol, 2020) is now cited in the introduction (p4) and discussed in more detail in the discussion p11.

9. Movies 5 and 6 are referred to in the text, but these are not in the files available or listed in the figure legends.

- We thank the reviewer for pointing out this mistake. We had removed these movies from the manuscript before submission, we have now deleted the corresponding references.

Reviewer #2 (Comments to the Authors (Required)):

Points addressed:

They do not even look at filopodia. The most critical gap is in connection with their conclusion that "βCdc42, embodies the major isoform regulating endocytosis" (from the Abstract). This is based only on analysis of macropinocytosis, a process in which Cdc42 is a secondary regulator. Yet they ignore the major Cdc42-dependent CLIC/GEEC pathway. A more careful examination of clathrin and clathrin-independent, Cdc42-dependent (CLIC) endocytosis is required to support this main conclusion.

- The goal of this study was not to elucidate the specific role of CDC42 isoforms in all the numerous CDC42 functions, but to investigate their specific contribution in directed migration of neural cells and explore the molecular basis of the isoform specificity.

Following the reviewer's suggestion we have now added data showing that filopodia formation, which is a classical CDC42-mediated process during migration, mainly involves CDC42u (Fig. S1E).

In this manuscript, we have looked at the role of CDC42 in pinocytosis during astrocyte migration and at the receptor recycling-dependent NPC chemotaxis, which has been shown to involve both clathrin dependent and clathrin-independent endocytosis. In these two circumstances, we show that CDC42b and N-WASP are major players. The literature indicate that N-WASP is involved in both clathrin-dependent endocytosis (Merrifield et al. Eur J Cell Biol. 2004) and in the CLIC/GEEC pathway (Chadda et al. Traffic, 2007), endocytosis. For these reasons, we would suggest that the specificity of CDC42b is related to both clathrin-dependent and -independent endocytosis. However, to take into account the reviewer's concern, we have more carefully worded our conclusions and now state that **CDC42b is the major isoform regulating N-WASP-mediated endocytosis**. The CI internalization pathways of macropinocytosis and phagocytosis are not discussed, primarily because they belong to a separate class of endocytic processes that involve internalization of relatively large (>1 μm) patches of membrane. However, these pathways may share some of the same molecular machinery, especially that related to the polymerization of actin in membrane remodelling.

A second gap is whether the functional difference between isoforms is due to differential activation in the different settings. Is the brain isoform poorly activated in the wounding assay, and conversely, is the placental isoform poorly activated in the endocytosis assay?

- The reviewer is asking a important and complex question. Are both isoforms simultaneously activated? We have previously shown the activation of CDC42 in the astrocyte wound healing assay (S. Etienne-Manneville, Cell, 2001). However, due to the lack of isoform-specific antibodies and the low expression of CDC42b, it is not possible to test the activity of each isoform separately.

Nevertheless, the fact that both isoforms change localization following wounding (CDC42u is recruited to the plasma membrane and CDC42b onto cytoplasmic vesicles) leads us to believe that both isoforms are activated. Moreover, pinocytosis which is CDC42b-dependent and Golgi reorientation which is CDC42u-dependent are both triggered by wounding, also suggesting that both isoforms are activated upon wounding. Similarly, during NPC chemotaxis, both isoforms are required for directed migration and yet, each is involved in specific functions with CDC42u controlling polarity and CDC42b controlling endocytosis. This shows that while both proteins are active, they control distinct cellular functions.

On the other hand, while both isoforms seem to be able to interact with the same GEFs and GAPs (from the mass spec data, Fig. 2), it is likely that their subcellular localization may affect their activity depending on the localization of GEFs or GAPs. Interestingly, GDI3, which is expressed in the brain (Adra et al. PNAS, 1997) localizes on the Golgi stacks (Zalcman et al. J Biol Chem, 1996; Brunet et al. Traffic, 2002) and co-localizes with CDC42b and controls CDC42b recruitment to cytoplasmic vesicles (our unpublished data). We are now studying the specific role of GDI3 in the regulation of CDC42-mediated function, but believe this is beyond the scope of this current manuscript. Because of the length limitation of the report format in J Cell Biol, we have not included this discussion in the manuscript.

Other issues:

Figure 2. what were the expression levels of the transfected GFP-Cdc42 constructs relative to endogenous and each other?

- Below is a blot showing the total cell lysate where we can pick up the endogenous protein (21kDa) vs the overexpressed protein GFP-CDC42. The signal of the endogenous protein is extremely low compared to the overexpressed one. Given the saturated GFP-CDC42 signal, it is not possible to quantify the relative level of the GFP tagged protein compared to the endogenous. The anti CDC42 antibody used here stains for both isoforms of CDC42.
- Since overexpression is too high compared to endogenous levels and distorts subcellular localization we chose to perform microinjection where we seem to catch early events of expression prior to overexpressing the protein. This led us to observe the differential sub cellular localization of both isoforms. This is also precisely why we performed different qPCR for comparing the expression of the both isoforms in different cell types.

Figure A: Westernblot anti-CDC42 showing the level of expression of the transfected GFP-tagged CDC42b and GFP-CDC42u in HEK cell total lysates. The endogenous CDC42 (including the CDC42b and CDC42u) are visible on a high exposure image.

Fig 3E and 4B. What were the expression levels of the rescue constructs relative to endogenous and each other?

- In these figures (now Fig 1H and 4C), the expression of proteins is done by microinjection into the nucleus of single cells and the analysis performed on GFP-

expression cells. This is done in the same conditions (4h after microinjection) than the localization experiments and thus also corresponds to a low level of expression of the proteins. However, it is not possible to analyse quantitatively the relative expression levels of the constructs.

Reviewer #3 (Comments to the Authors (Required)):

Points

1. The naming of the two Cdc42 isoforms is confusing, especially because plaCdc42 is not specific to the placenta, and the authors variably use capitals and small letters and sometimes use PlCdc42. In addition, splice variant information is always placed after the gene name (e.g. Rac1b) rather than before (as done here). The Cerione group have just published in JBC using the nomenclature Cdc42u (ubiquitous) and Cdc42b (brain). It would be ideal if the same nomenclature is used here, consistent with them, rather than introducing a different nomenclature. The Cerione group paper should also be added to the Discussion here because it is highly relevant, although the analysis carried out is very different to that described here and not overlapping.

➤ We agree with the reviewers with regard to nomenclature. Following their recommendation and the nomenclature used by Endo et al, JBC, 2020, we have changed the text as follows:

Brain Cdc42 = brCdc42 to **CDC42b**

Placental Cdc42 = plaCdc42 (placental) to **CDC42u** (ubiquitous).

➤ The Cerione group paper (Endo et al., J Cell Biol, 2020) is now cited and discussed in the introduction and the discussion of the revised manuscript.

2. For brCdc42 (Fig. 1A), is it known if the CAAX box is not processed by removal of AAX followed by methylation? If it were processed, then palmitoylation would not occur. Is removal of the AAX regulatable?

➤ Nishimura and colleagues (cited p7 and p8 of the manuscript) have showed that Cdc42b undergoes two different types of posttranslational modification: classical CaaX processing or novel tandem prenylation and palmitoylation at the CCaX cysteines. In the dual lipidation pathway, Cdc42b is first prenylated, but then bypasses proteolysis and carboxymethylation to allow the addition of the palmitate at the second cysteine. This mechanism applies to proteins ending with a CCaX motif like Ra1A, Ra1B and PRL-3 in addition to Cdc42b (Falsetti et al. Mol Cell Biol 2007, Zeng et al. J. Biol. Chem. 2000, Nishimura et al. Mol. Cell. Biol., 2013).

3. Movies: for plaCdc42 there appears to be a lot of dynamic 'vesicular' movement, and it is not uniformly 'cytoplasmic' as the authors state in the text (movies 1 and 3). Some of it appears to overlap with brCdc42 (Movie 1). The text should describe exactly what is observed in the movies, unless the movie is unrepresentative of what they have found in multiple movies (and therefore should be replaced). By contrast brCdc42 'vesicles' are surprising static (movies 1 and 2) for the most part - this might be because overexpression impairs vesicle movement. There is also a concern that the fusion with GFP alters their ability to signal/localize correctly. What is the localization with a smaller, different tag?

➤ We have changed the movie 1 which was, indeed, difficult to interpret. The new movie 1 clearly shows that CDC42b accumulates on intracellular vesicles, much more than CDC42u. However, the reviewer is right in saying that CDC42u can sometimes be seen

on some vesicles which are forming close to the leading edge plasma membrane (as observed in movie 2). We think that this localization is probably related to the minor, but not null, role of CDC42u in macropinocytosis (Fig. 4A).

Looking at many movies, the movement of CDC42b positive vesicles appears to depend on the size of the vesicles. Although we have not quantified it, large vesicles tend to move much less than small ones. As the reviewer suggests, we also suspect that the level of expression of CDC42b (or the length of time during which it is expressed following microinjection) parallels with an increase in the vesicle number and size and a decrease in their motility.

- We have used a smaller tag, the ALFA tag, and confirmed the specific localization of the two isoforms (in Fig. S1F).

4. Fig. 1C/D: how many cells were quantified in total and in each of the individual experiments? What is the definition of 'strong co-localization' with GM130/EEA1? Pearson's correlation coefficient should be used for both GM130 and EEA1 graphs to be consistent. What is the correlation coefficient for plaCdc42 and brCdc42? There are some fiber-like structures running across the images in Fig. 1C/D where they both appear to co-localize. What are these and are they observed in other images?

- Fig 1C,D now Fig 3B,C. The number of cells per independent experiments and the number of independent experiments is indicated in the corresponding figure legend.

We now represent the Pearson's correlation coefficient of CDC42u and CDC42b for both GM130 and EEA1 colocalization. In this case, we show the values and means \pm SEM of 3 independent experiments and at least 30 cells were analysed per condition.

- CDC42 is an actin regulatory protein the fiber like structures are actin filaments running across the cell and in filopodial projections. Such localization of CDC42 has been shown in the past in several studies using different cell types (For instance : Krugman et al. *Curr Biol*, 2001, Cau et al. *J Cell Sci*, 2005)

5. Fig. 1E: Why does brCdc42 without geranylgeranylation not go to the nucleus? What happens if both cysteines are mutated - does it then go to the nucleus? Can it be palmitoylated without prenylation (like RhoU/V)?

- The reviewer is right in noticing that inhibition of prenylation leads to the accumulation of the unprenylated u(SVLL) CDC42 in the cell nucleus. This nuclear accumulation has been observed for other unprenylated mutants of Rho GTPases such as Rac and RhoA (Michaelson et al. 2008; Abdrabou and Wang 2018). We ran an in silico NLS prediction for Cdc42 variants using cNLS Mapper. Both Cdc42 variants were predicted to encode a bipartite NLS encoded by 28 and 29 C-ter amino acids for Cdc42u and Cdc42b respectively (Fig B below). However, Cdc42 NLS is only present in the absence of the C-ter prenylation, which perturbs nuclear import. This type of NLS is referred to as a cryptic NLS, which is also observed in the case of Rac1 (Michaelson et al. 2008).

Despite similar cryptic NLS regions, the b(SCIF) and the b(CSIF) CDC42 mutants do not accumulate in the nucleus. The double lipid mutant b(SSIF) does (although less than u(SVLL)) (Fig C below). One hypothesis is that a single mutation in the CCIF sequence does not totally abrogate the addition of the other lipid anchors. It is certainly true for the b(CSIF) mutant which can still prenylated. It is however less clear for the b(SCIF) mutant which, in principle cannot be palmitoylated. An alternative hypothesis is that the differences in the polybasic residue region (PBR) of the two CDC42 isoforms influence the nuclear translocation. Increased PBR region (as is the case between

CDC42b and CDC42u) is associated with an increased nuclear accumulation in the case of several Rho GTPases and their isoforms (Lanning et al. J Biol Chem, 2004).

Figure B NLS prediction was performed for i) Cdc42u and ii) Cdc42b using cNLS Mapper, available at <http://nls-mapper.iab.keio.ac.jp/>. Both proteins have their predicted NLS situated in their c-ter comprising of 28 and 29 amino acids highlighted in red for Cdc42u and Cdc42b respectively. They both lack a monopartite NLS and their respective scores for a bipartite NLS are represented.

Figure C Representing all the Cdc42 mutants generated by eliminating lipid modifications with Cys to Ser point mutations in both isoforms.

6. Fig. 2B: There is one protein outlier that binds much more to plaCdc42 than BrCdc42. Which protein is this – if it is Cdc42, then why was it not detected for brCdc42? It should be described in the results/labelled on the figure. Network analysis of the proteins identified would really enhance these important data - albeit not so relevant to the differences between the isoforms, the information of Cdc42-interacting proteins/complexes is still useful.

- Changes have been made to the mass spec plots and suppressed the non-relevant interactome (CRAPOME, see our answer to point 1 of reviewer 1 for details). All interactors have now been labelled for better understanding. Cdc42 interactors have now been categorised into GEFs, GAPs and effectors (revised Fig. 2D-F).

7. Fig. S1E - it would be useful to show the knockdown of brCdc42 on the mRNA level for comparison with plaCdc42, so that the relative knockdown of each isoform can be directly compared.

- The direct comparison of the effect of CDC42b and CDC42u knockdown on mRNA level is now shown In Fig S1D.

8. Fig. 3: For the graphs, all the data should be provided on the plot i.e. show a dot for each experiment, and also add graphs to show the range of data for all the cells tracked/analysed (dot plots). The total number of cells tracked or analysed should be given in the figure legend. Finally, absolute p values should be shown above each bar not asterisks (including those with no asterisk) - this is much preferred now. Please substitute the Greek letter for 'zeta' in the figure legends and figure.

- Following the reviewer's and JCB recommendations (Lord et al. J Cell Biol, 2020), we now show the data points of each experiment together with the mean and SEM of these experiments. The number of cells taken into account for each experiment is indicated in the figure legend. The asterisks have been replaced by the exact p values in all figures.
- PKCzeta is now PKCζ (now in revised Fig. 1F, 1G)

9. Fig. 4: as for Fig. 3, give absolute p values and total numbers of cells analysed, adding graphs to show the variability in dextran uptake across all the cells analysed.

- Following the reviewer's and JCB recommendations (Lord et al. J Cell Biol, 2020), we now show the data points of each experiment together with the mean and SEM of these experiments. The number of cells taken into account for each experiment is indicated in the figure legend. The asterisks have been replaced by the exact p values in all figures.

10. Figure S2: given that WASL is N-WASP, it is important to state this in the final column of the table, where it would be more helpful to give the abbreviated name rather than writing out the full name. This is also relevant for some other targets e.g., PAR6. Please use Greek letters in this column where relevant.

- To avoid the inconsistencies that the reviewer highlights we now use uniform nomenclature for all proteins described in the paper and not gene names in the proteomic screen. Greek letters have been replaced where necessary both for protein names in FigS2 and in the main text.

11. Fig. 5 and Fig S3: Amend graphs and figure legend information as for Fig. 3.

- This has been done. See answer to comments 8 and 9.

12. There are a few typos.

- We have made major modifications in the text and hope that we have corrected all the typos.

August 25, 2023

RE: JCB Manuscript #202004092R-A

Dr. Sandrine Etienne-Manneville
Institut Pasteur
Institut Pasteur
25 rue du Dr Roux
Paris 75015
France

Dear Sandrine,

Thank you for submitting your revised manuscript entitled "The distinct localization of CDC42 isoforms is responsible for their specific functions during migration". We would be happy to publish your paper in JCB pending completely addressing the remaining reviewer comments with textual revisions as well as final revisions necessary to meet our formatting guidelines (see details below)

A. MANUSCRIPT ORGANIZATION AND FORMATTING:

- 1) Text limits: Character count for Reports is < 20,000, not including spaces. Count includes abstract, introduction, * combined results and discussion, and acknowledgments. Count does not include title page, figure legends, materials and methods, references, tables, or supplemental legends.
- 2) Figures limits: Reports may have up to 5 main text figures.
- 3) * Figure formatting: Scale bars must be present on all microscopy images, including inset magnifications. * Molecular weight or nucleic acid size markers must be included on all gel electrophoresis. The use of red/green should be avoided when possible.
*
- 4) Statistical analysis: Error bars on graphic representations of numerical data must be clearly described in the figure legend. The number of independent data points (n) represented in a graph must be indicated in the legend. Statistical methods should be explained in full in the materials and methods. For figures presenting pooled data the statistical measure should be defined in the figure legends. Please also be sure to indicate the statistical tests used in each of your experiments (either in the figure legend itself or in a separate methods section) as well as the parameters of the test (for example, if you ran a t-test, please indicate if it was one- or two-sided, etc.). Also, if you used parametric tests, please indicate if the data distribution was tested for normality (and if so, how). If not, you must state something to the effect that "Data distribution was assumed to be normal but this was not formally tested."
- 5) Abstract and title: The abstract should be no longer than 160 words and should communicate the significance of the paper for a general audience. The title should be less than 100 characters including spaces. Make the title concise but accessible to a general readership.
- 6) Materials and methods: Should be comprehensive and not simply reference a previous publication for details on how an experiment was performed. Please provide full descriptions in the text for readers who may not have access to referenced manuscripts.
- 7) Please be sure to provide the sequences for all of your primers/oligos and RNAi constructs in the materials and methods. You must also indicate in the methods the source, species, and catalog numbers (where appropriate) for all of your antibodies. Please also indicate the acquisition and quantification methods for immunoblotting/western blots.
- 8) Microscope image acquisition: The following information must be provided about the acquisition and processing of images:
 - a. Make and model of microscope
 - b. Type, magnification, and numerical aperture of the objective lenses
 - c. Temperature
 - d. Imaging medium

- e. Fluorochromes
- f. Camera make and model
- g. Acquisition software
- h. Any software used for image processing subsequent to data acquisition. Please include details and types of operations involved (e.g., type of deconvolution, 3D reconstitutions, surface or volume rendering, gamma adjustments, etc.).

10) Supplemental materials: There are strict limits on the allowable amount of supplemental data. Reports may have up to 3 supplemental figures. Please also note that tables, like figures, should be provided as individual, editable files. A summary of all supplemental material should appear at the end of the Materials and methods section.

13) ORCID IDs: ORCID IDs are unique identifiers allowing researchers to create a record of their various scholarly contributions in a single place. Please note that ORCID IDs are now *required* for all authors. At resubmission of your final files, please be sure to provide your ORCID ID and those of all co-authors.

Please note that JCB now requires authors to submit Source Data used to generate figures containing gels and Western blots with all revised manuscripts. This Source Data consists of fully uncropped and unprocessed images for each gel/blot displayed in the main and supplemental figures. Since your paper includes cropped gel and/or blot images, please be sure to provide one Source Data file for each figure that contains gels and/or blots along with your revised manuscript files. File names for Source Data figures should be alphanumeric without any spaces or special characters (i.e., SourceDataF#, where F# refers to the associated main figure number or SourceDataFS# for those associated with Supplementary figures). The lanes of the gels/blots should be labeled as they are in the associated figure, the place where cropping was applied should be marked (with a box), and molecular weight/size standards should be labeled wherever possible.

Journal of Cell Biology now requires a data availability statement for all research article submissions. These statements will be published in the article directly above the Acknowledgments. The statement should address all data underlying the research presented in the manuscript. Please visit the JCB instructions for authors for guidelines and examples of statements at (<https://rupress.org/jcb/pages/editorial-policies#data-availability-statement>).

B. FINAL FILES:

****It is JCB policy that if requested, original data images must be made available to the editors. Failure to provide original images upon request will result in unavoidable delays in publication. Please ensure that you have access to all original data images prior to final submission.****

****The license to publish form must be signed before your manuscript can be sent to production. A link to the electronic license to publish form will be sent to the corresponding author only. Please take a moment to check your funder requirements before choosing the appropriate license.****

Thank you for this interesting contribution, we look forward to publishing your paper in Journal of Cell Biology.

Sincerely,

Ian Macara, Ph.D.
Editor

Andrea L. Marat, Ph.D.
Senior Scientific Editor

Journal of Cell Biology

Reviewer #1 (Comments to the Authors (Required)):

The manuscript by Ravichandran, Hanisch et al addresses most of my previous concerns.

A couple of minor things that the authors should to address below:

-Fig 1A-1C: no change to migration velocity is mentioned, but this data is missing.

-In the next describing Fig 2, the authors state "The GFP-tagged WT- pr CA-Cdc42 resins...". No data is shown for WT in the mass Spec, only the CA mutant." Either show this data (and therefore re-review is needed) or remove the 'WT'.

-In the text describing Fig 3D, the authors state that the CSIF mutation "inhibits recruitment of CDC42b to the plasma membrane and on endocytic vesicles, but did not prevent its association with the Golgi apparatus (Fig. 3D)". No such co-localisations are shown, and therefore this statement needs to describe what is actually there.

Reviewer #2 (Comments to the Authors (Required)):

The manuscript from Ravichandran et al is somewhat improved, though most of my requests have been deflected as too difficult or "beyond the scope". It remains a basically sound paper that convincingly shows that Cdc42 isoforms that differ in C-terminal lipid modifications also differ in localization and effector pathways in vivo, both of which contribute to cell migration. However, it is still lacking deep mechanistic insights. I think it is basically publishable but borderline for JCB.

Minor criticism:

In Fig 4F, the co-localization of Cdc42 isoforms with N-Wasp needs to be quantified.

Reviewer #3 (Comments to the Authors (Required)):

This revised manuscript is much improved and addresses nearly all of my comments concerning the original manuscript (submitted 2020) with revisions to the text and figures. There are a few revisions required to the revised manuscript, which I relate back to my points in previous review:

1. Naming of the Cdc42 isoforms. Thank you for changing to Cdc42b and Cdc42u in the text and figures. Please also change the movie labeling: the movies still have brCdc42 and PICd42 on them. Reference to Endo et al., 2020: this is quoted in the Discussion, but the reference has not been inserted in the Introduction. Please insert there too.
2. Please state specifically in the text that Cdc42b does not always undergo proteolysis and methylation, and hence can be palmitoylated, when referring to the Nishimura paper (note on p. 7, Nishimura et al has not been inserted correctly).
5. Please add a representative double Cdc42b-SSIF mutant image to the relevant figure. Localization of each mutant should be quantified by measuring the nuclear: cytoplasmic ratio. The authors cannot be sure that, under their conditions, GGTI treatment of Cdc42b-expressing cells or the Cdc42b-SCIF mutant does not permit palmitoylation, since RhoU and RhoV C-termini can be palmitoylated (but are not prenylated). The text should be altered so that it is not so definitive at the bottom of p. 7 and top of p. 8 (and Figure S3 legend). For example, GGTI only inhibits geranyl geranylation and they have not proved that GGTI also prevents palmitoylation of Cdc42b. This has only been reported in one publication, as far as I am aware.
7. mRNA levels following siRNA knockdown of Cdc42b and Cdc42u. The authors state in their rebuttal that the data showing direct comparisons between Cdc42b and Cdc42u mRNA levels is shown in Figure S1D. However, Figure S1D does not show this. Figure S1A shows knockdown of Cdc42u but not Cdc42b. This figure should be modified to include the data using the Cdc42b siRNAs and subsequent levels of Cdc42b versus Cdc42u mRNAs.
12. The text and figure legend revisions have incorporated new typos and grammatical errors (and there are still some old ones) so need careful editing.

INSTITUT PASTEUR

Cell Polarity, Migration and Cancer

From: *Etienne-Manneville Sandrine*

Téléphone: 01 40 61 39 05

Télécopie: 01 45 68 85 48

E-mail: *setienne@pasteur.fr*

Paris, 2023 September 20

SUBJECT: JCB manuscript #202004092R-A

Dear Andrea, Dear Ian
Dear editor,

Thank you very much for your consideration of our manuscript "The distinct localization of CDC42 isoforms is responsible for their specific functions during migration". We were pleased to learn that it is now accepted in the Journal of Cell Biology.

We are now submitting the re-revised manuscript that includes the final revisions necessary to meet the JCB formatting guidelines as well as the answers to the latest comments of the reviewers (see our point-by-point answers below)

Thank you very much, and I look forward to read our manuscript in the Journal of Cell Biology

Best regards,

Sandrine Etienne-Manneville, PhD
Cell Polarity, Migration and Cancer Lab, Institut Pasteur-CNRS

Point-by-point answers to the reviewers

Reviewer #1 (Comments to the Authors (Required)):

The manuscript by Ravichandran, Hanisch et al addresses most of my previous concerns. A couple of minor things that the authors should to address below:

- Fig 1A-1C: no change to migration velocity is mentioned, but this data is missing.
The text for migration velocity has been removed as polarity is being assessed here

- In the next describing Fig 2, the authors state "The GFP-tagged WT- pr CA-Cdc42 resins...". No data is shown for WT in the mass Spec, only the CA mutant." Either show this data (and therefore re-review is needed) or remove the 'WT'.
WT has been removed in the text it was present in the older version of the article

-In the text describing Fig 3D, the authors state that the CSIF mutation "inhibits recruitment of CDC42b to the plasma membrane and on endocytic vesicles, but did not prevent its association with the Golgi apparatus (Fig. 3D)". No such co-localisations are shown, and therefore this statement needs to describe what is actually there.
Extra text removed and only what is observed is described

Reviewer #2 (Comments to the Authors (Required)):

The manuscript from Ravichandran et al is somewhat improved, though most of my requests have been deflected as too difficult or "beyond the scope". It remains a basically sound paper that convincingly shows that Cdc42 isoforms that differ in C-terminal lipid modifications also differ in localization and effector pathways in vivo, both of which contribute to cell migration. However, it is still lacking deep mechanistic insights. I think it is basically publishable but borderline for JCB.

Minor criticism:

In Fig 4F, the co-localization of Cdc42 isoforms with N-Wasp needs to be quantified.

We have added a plot profile that shows the fluorescence intensity profile of the two proteins

Reviewer #3 (Comments to the Authors (Required)):

This revised manuscript is much improved and addresses nearly all of my comments concerning the original manuscript (submitted 2020) with revisions to the text and figures. There are a few revisions required to the revised manuscript, which I relate back to my points in previous review:

1. Naming of the Cdc42 isoforms. Thank you for changing to Cdc42b and Cdc42u in the text and figures. Please also change the movie labeling: the movies still have brCdc42 and PICd42 on them. *Changed*
Reference to Endo et al., 2020: this is quoted in the Discussion, but the reference has not been inserted in the Introduction. Please insert there too. *Inserted*

3. Please state specifically in the text that Cdc42b does not always undergo proteolysis and methylation, and hence can be palmitoylated, when referring to the Nishimura paper (note on p. 7, Nishimura et al has not been inserted correctly). *done*

5. Please add a representative double Cdc42b-SSIF mutant image to the relevant figure. Localization of each mutant should be quantified by measuring the nuclear: cytoplasmic ratio.

Changes made to Fig.3D

The authors cannot be sure that, under their conditions, GGTI treatment of Cdc42b-expressing cells or the Cdc42b-SCIF mutant does not permit palmitoylation, since RhoU and RhoV C-termini can be palmitoylated (but are not prenylated). The text should be altered so that it is not so definitive at the bottom of p. 7 and top of p. 8 (and Figure S3 legend). For example, GGTI only inhibits geranyl geranylation and they have not proved that GGTI also prevents palmitoylation of Cdc42b. This has only been reported in one publication, as far as I am aware. *Text edits done*

7. mRNA levels following siRNA knockdown of Cdc42b and Cdc42u. The authors state in their rebuttal that the data showing direct comparisons between Cdc42b and Cdc42u mRNA levels is shown in Figure S1D. However, Figure S1D does not show this. Figure S1A shows knockdown of Cdc42u but not Cdc42b. This figure should be modified to include the data using the Cdc42b siRNAs and subsequent levels of Cdc42b versus Cdc42u mRNAs. *Figure S1A now shows the effect of each specific siRNA on CDC42u mRNA level and Figure S1B shows the effect of each specific siRNA on the total CDC42 mRNA. From this, it is possible to determine the levels of CDC42b in each conditions. We have corrected the figure legend accordingly.*

12. The text and figure legend revisions have incorporated new typos and grammatical errors (and there are still some old ones) so need careful editing.

We have carefully read the manuscript and corrected some typos.